# Kinase-dead ATM protein is highly oncogenic and can be preferentially targeted by Topo-isomerase I inhibitors

Kenta Yamamoto[1,2,3,4], Jiguang Wang[5,6,7], Lisa Sprinzen[1,2,3,4], Jun Xu[8], Christopher J Haddock[9], Chen Li[1,2,3], Brian J Lee[1,2,3], Denis G Loredan[1,2,3], Wenxia Jiang[1,2,3], Alessandro Vindigni[9], Dong Wang[8], Raul Rabadan[5,6,7], Shan Zha[1,2,3,10,11,12]*

[1]Institute for Cancer Genetics, Columbia Unviersity, New York, United States; [2]Department of Pathology and Cell Biology, Columbia University, New York, United States; [3]College of Physicians and Surgeons, Columbia University, New York, United States; [4]Pathobiology and Molecular Medicine Graduate Program, Columbia University, New York, United States; [5]Department of Biomedical Informatics, Columbia University, New York, United States; [6]Department of Systems Biology, Columbia University, New York, United States; [7]College of Physicians & Surgeons, Columbia University, New York, United States; [8]Skaggs School of Pharmacy & Pharmaceutical Sciences, University of California San Diego, La Jolla, United States; [9]Edward A Doisy Department of Biochemistry and Molecular Biology, Saint Louis University School of Medicine, St. Louis, United States; [10]Division of Pediatric Oncology, Hematology and Stem Cell Transplantation, Columbia University, New York, United States; [11]Department of Pediatrics, Columbia University, New York, United States; [12]College of Physicians & Surgeons, Columbia University, New York, United States

*For correspondence: sz2296@cumc.columbia.edu

Competing interests: The authors declare that no competing interests exist.

**Abstract** Missense mutations in ATM kinase, a master regulator of DNA damage responses, are found in many cancers, but their impact on ATM function and implications for cancer therapy are largely unknown. Here we report that 72% of cancer-associated ATM mutations are missense mutations that are enriched around the kinase domain. Expression of k̲inase-d̲ead ATM ($Atm^{KD/-}$) is more oncogenic than loss of ATM ($Atm^{-/-}$) in mouse models, leading to earlier and more frequent lymphomas with Pten deletions. Kinase-dead ATM protein (Atm-KD), but not loss of ATM (Atm-null), prevents replication-dependent removal of Topo-isomerase I-DNA adducts at the step of strand cleavage, leading to severe genomic instability and hypersensitivity to Topo-isomerase I inhibitors. Correspondingly, Topo-isomerase I inhibitors effectively and preferentially eliminate $Atm^{KD/-}$, but not Atm-proficient or $Atm^{-/-}$ leukemia in animal models. These findings identify ATM kinase-domain missense mutations as a potent oncogenic event and a biomarker for Topo-isomerase I inhibitor based therapy.

## Introduction

ATM kinase is a tumor suppressor that has a central role in the DNA damage responses. Germline inactivation of ATM causes ataxia-telangiectasia (A-T), which is associated with greatly increased risk of lymphoma and leukemia (*Lavin, 2008*). Somatic mutations of ATM occur frequently in mantle cell lymphomas (MCL), chronic lymphoblastic leukemia (CLL) and T-cell prolymphocytic leukemia

**eLife digest** Cancer is a genetic disease. To remain healthy, therefore, it is essential that cells do not accrue too many dangerous mutations in their DNA that allow cancers to grow and develop. An enzyme called ATM helps to do just that. DNA damage activates ATM, which, in turn, adds phosphate groups to other proteins. These newly tagged proteins then stop cells dividing until the DNA has been repaired.

Human cancers often switch off ATM, either by completely deleting the enzyme or mutating it. This renders ATM unable to add phosphate groups to proteins, and so allows the cancer cells to continue proliferating even in the face of DNA damage.

Yamamoto et al. wanted to know whether cancers that completely lack ATM behave differently from cancers that contain an inactive version of the enzyme. Studying mice that were engineered to have an inactive version of ATM in their blood cells showed that such mice developed blood cancers faster than mice have no ATM in their blood cells. In particular, cancer cells with the inactive form of the ATM enzyme accumulated more DNA damage than cells that lacked the enzyme completely.

Using biochemical techniques, Yamamoto et al. then showed that the inactive form of ATM can prevent other enzymes from repairing DNA. Drugs that inhibit one of these repair enzymes – called topo-isomerase I – are already used in cancer treatments. These drugs were particularly effective on tumors with the inactive version of the ATM enzyme.

As ATM is commonly mutated in human cancers, the next steps that follow on from this research are to develop methods to test which cancers contain the inactive form of the ATM enzyme. Clinical trials could then investigate how effectively topo-isomerase I inhibitors treat these specific types of cancer.

(TPLL) (*Wang et al., 2011*; *Beà et al., 2013*; *Stilgenbauer et al., 1997*). In lymphomas, *ATM* mutations often occur with concurrent heterozygous deletion of 11q23 including *ATM* (*Wang et al., 2011*; *Beà et al., 2013*; *Stilgenbauer et al., 1997*), potentially leading to the expression of mutated ATM in the absence of the wild-type (WT) protein. Recent sequencing studies also identified recurrent ATM mutations in 2–8% of breast, pancreas or gastric cancers (*Roberts et al., 2012*; *Cremona and Behrens, 2014*). While the majority of A-T patients (~90%) have *truncating ATM* mutations that result in little or no ATM protein expression (*Concannon and Gatti, 1997*), missense *ATM* mutations are more common in cancers and with the exception of the few that cause A-T, their biological functions are unknown.

As a serine/threonine protein kinase, ATM is recruited and activated by DNA double strand breaks (DSBs) through direct interactions with the MRE11, RAD50 and NBS1 (MRN) complex (*Lee and Paull, 2004*; *Paull, 2015*; *Stewart et al., 1999*; *Carney et al., 1998*). Activated ATM phosphorylates >800 substrates implicated in cell cycle checkpoints, DNA repair, and apoptosis to suppress genomic instability and tumorigenesis. ATM activation is also associated with intermolecular autophosphorylation (*Bakkenist and Kastan, 2003*; *Kozlov et al., 2011*). Studies in human cells suggest that auto-phosphorylation is required for ATM activation (*Bakkenist and Kastan, 2003*; *Kozlov et al., 2011*). However, alanine substitutions at one or several auto-phosphorylation sites do not measurably affect ATM kinase activity in transgenic mouse models (*Daniel et al., 2008*; *Pellegrini et al., 2006*), leaving the biological function of ATM auto-phosphorylation unclear. In this context, we and others generated mouse models expressing kinase dead (KD) ATM protein (Atm-KD) (*Yamamoto et al., 2012*; *Daniel et al., 2012*). In contrast to the normal development of $Atm^{-/-}$ mice, $Atm^{KD/-}$ and $Atm^{KD/KD}$ mice die during early embryonic development with severe genomic instability (*Yamamoto et al., 2012*; *Daniel et al., 2012*; *Barlow et al., 1996*), implying that the expression of Atm-KD in the absence of WT ATM further inhibits DNA repair beyond the loss of ATM. Among the major DNA DSB repair pathways, non-homologous end-joining (NHEJ) is uniquely required for the repair phases of lymphocyte specific V(D)J recombination and class switch recombination (CSR). ATM promotes NHEJ (*Zha et al., 2011b*; *Bredemeyer et al., 2006*), and $Atm^{-/-}$ mice and A-T patients suffer from primary immunodeficiency (*Jiang et al., 2015b*; *Zha et al., 2011a*). The end-joining defects in V(D)J recombination and CSR are similar in $Atm^{KD/-}$ and $Atm^{-/-}$ lymphocytes,

suggesting that Atm-KD protein is not dominant-inhibitory for NHEJ. Meanwhile, $Atm^{KD/-}$ cells show moderate yet significant hypersensitivity to PARP inhibitors in comparison to both $Atm^{-/-}$ and $Atm^{+/+}$ cells (*Daniel et al., 2012*), and ATM kinase inhibitor, but not loss-of-ATM, reduced homologous recombination (HR) as measured by sister chromatid exchange (SCE) (*White et al., 2010*) and the DR-GFP HR reporter (*Rass et al., 2013*; *Kass et al., 2013*), suggesting a role of ATM auto-phosphorylation in replication associated homology dependent repair. Yet, it is unknown how ATM auto-phosphorylation contributes to DNA replication and HR beyond its previously identified signaling roles. Finally, consistent with the 'inter'-molecular autophosphorylation model, $Atm^{+/KD}$ mice are largely normal (*Yamamoto et al., 2012*), suggesting that $ATM^{KD}$ mutation carriers could be asymptomatic and somatic loss of the WT allele in the carriers might create the Mut/Del status reported in human cancers and lymphomas.

Here we report that 72% of human cancer-associated ATM mutations are missense mutations that are highly enriched in the kinase domain. We further show that conditional expression of Atm-KD protein alone (somatic inactivation of the conditional allele ($Atm^C$) in $Atm^{KD/C}$ mice to generate the $Atm^{KD/-}$ cells) in murine hematopoietic stem cells (HSCs) is more oncogenic than the complete loss of ATM ($Atm^{-/-}$). $Atm^{KD/-}$ cells are selectively hypersensitive to Topo Isomerase I (Topo1) inhibitors, in part because the Atm-KD protein physically blocks replication-dependent strand cleavage upon Topo1 inhibition. Correspondingly, Topo1 inhibitor selectively eradicates Notch1-driven $Atm^{KD/-}$ leukemia, but not the isogenic parental Atm proficient leukemia, identifying Topo1 inhibitors as a *targeted* therapy for human cancers carrying missense ATM kinase domain mutations.

## Results

### Cancer-associated ATM mutations are enriched for kinase domain missense mutations

Among the 5402 cases in The Cancer Genome Atlas (TCGA), we identified 286 unique non-synonymous mutations of *ATM*. While truncating (nonsense/frameshift) mutations compose 83% (373/447) of A-T associated point mutations (>1000 patients), 72% (206/286) of non-synonymous point mutations of *ATM* in TCGA are missense mutations (*Figure 1A*, *Supplementary file 1A,B*). Permutation analyses show that *ATM* gene is not hyper-mutated, but the kinase-domain is mutated 2.5 fold more frequently than otherwise expected in TCGA (*Figure 1—figure supplement 1A*, p<0.01). The mutation density calculated using the Gaussian Kernel model revealed that cancer associated missense *ATM* mutations in TCGA cluster around the C-terminal kinase domain, while truncating mutations (in A-T or TCGA) span the entire ATM protein (*Figure 1B* and *Figure 1—figure supplement 1B*). Given the severe phenotype of $Atm^{KD/-}$, but not $Atm^{KD/+}$ cells, we further analyzed the subset (~105/286) of *ATM* missense mutations in TCGA that are concurrent with heterozygous loss of *ATM* (shallow deletion) or truncating mutations in the same case, and found that, again, *ATM* missense mutations cluster around the C-terminal kinase domain even in this smaller subset (*Figure 1B*). The kinase and FATC domains of ATM share 31% sequence identity with mTOR, a related phosphatidylinositol 3-kinase-related protein kinase (PIKK) for which the high resolution crystal structure is available (*Yang et al., 2013*). Homology modeling using mTOR (PDB 4JSP) (*Yang et al., 2013*) revealed that 64% (27/42) (at 18 unique amino acids) of ATM kinase domain missense mutations from TCGA, affect highly conserved residues and 50% (21/42) of the mutations (red on the ribbon structure) likely abolish kinase activity based on structural analyses (*Figure 1C*, *Figure 1—figure supplement 1C*). Specifically, residues K2717, D2720, H2872, D2870, N2875 and D2889 of human ATM are predicted to bind ATP or the essential $Mg^+$ ion (*Figure 1—figure supplement 1D*). Notably, N2875 is mutated in two TCGA cases at the time of initial analyses. One of the two cases have concurrent shallow deletion in this region (*Supplementary file 1B*). Since then, one additional N2875 mutation was reported in a prostate cancer case (TCGA-YL-A8S9) with an allele frequency of 0.92, consistent with homozygosity. Mutations corresponding to N2875K of human ATM were previously engineered into the $Atm^{KD/+}$ mice together with the corresponding D2870A mutation of human ATM. This combination was found to abolish ATM kinase activity without significantly affecting ATM protein levels (*Yamamoto et al., 2012*; *Canman, 1998*). Finally, immunoblotting confirmed the expression of catalytically inactive ATM protein in several human cancer cell lines with missense mutations around the kinase domain (CCLE, Broad Institute) (*Figure 1—figure supplement 1E–F*).

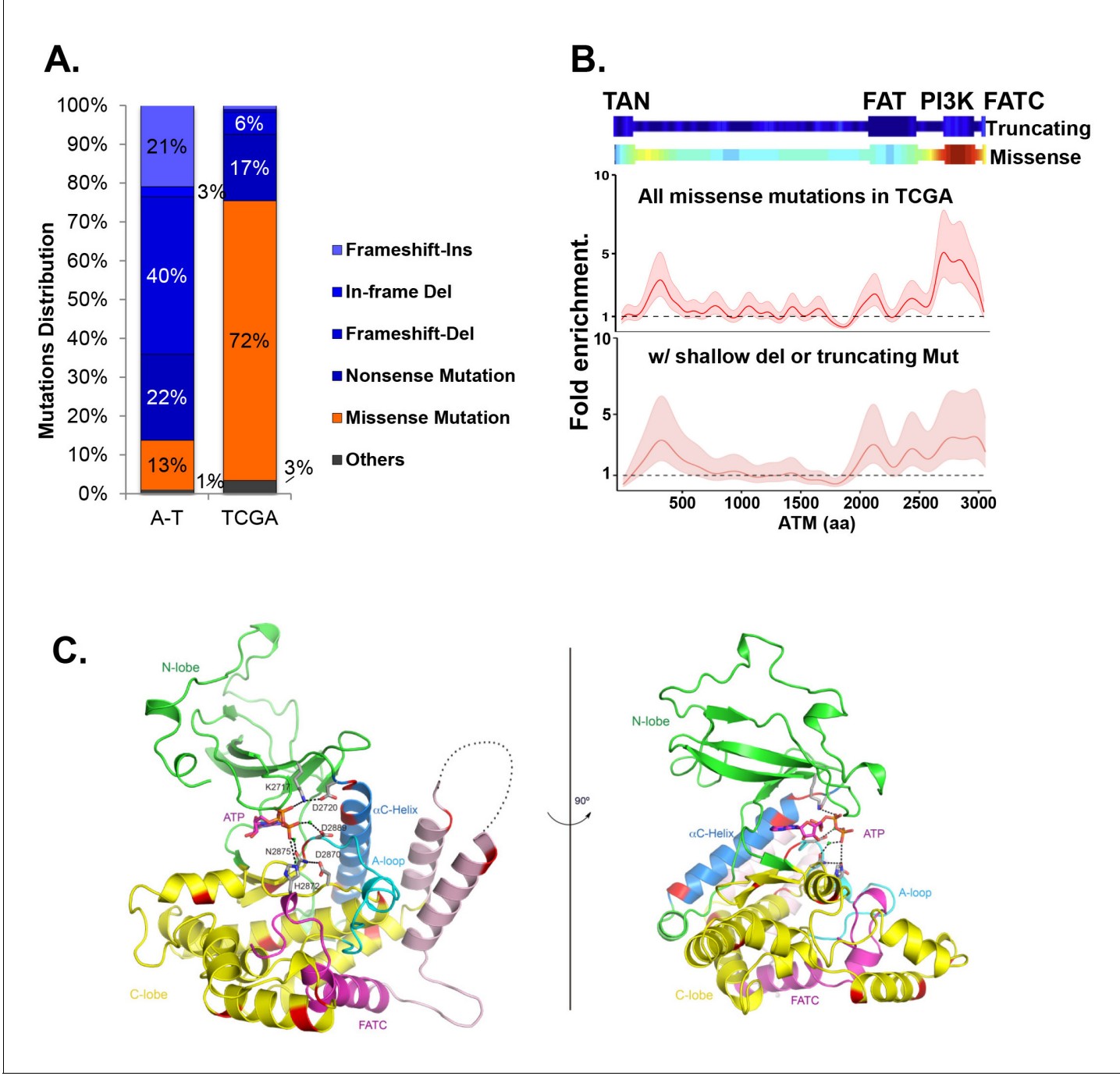

**Figure 1.** Cancer-associated ATM mutations are enriched for kinase domain missense mutations. (**A**) Classification of unique and nonsynonymous mutations reported for A-T patients (n > 1000) in the Leiden open variable database (http://chromium.lovd.nl/LOVD2/home.php?select_db=ATM) (**Supplementary file 1A**) or unique mutations from 5402 cancer cases in TCGA (**Supplementary file 1B**). 'Others' includes splice-site and stop codon mutations. (**B**) Heatmaps for the frequency of cancer-associated truncating mutations (upper panel) or nonsynonymous substitutions (missense) mutations (lower panel) along the human ATM polypeptide obtained using Gaussian kernel based analyses (see Materials and methods). Warmer colors indicate denser mutation distribution. The shadowed curves in the middle and the bottom panel reflect the ratio of actual over expected mutation frequencies, calculated by Gaussian kernel method in all TCGA ATM missense mutations (middle) or only missense mutations with concurrent shallow deletion/truncating mutations (bottom). The confidence intervals (shadows) are estimated by fitting a binomial distribution (see Materials and methods). The dashed line at 1 represents the expected rate of mutations. The x-axis is the amino acid number (from 1–3057, including stop code) along the human ATM protein. (**C**) Homology model of the human ATM kinase domain is shown in two views rotated by 90°. The N-lobe (aa 2640–2774) is shown in green; the C-lobe (aa 2775–3057,) is shown in yellow; the activation loop (aa 2883–2907) is shown in cyan; the FATC domain (aa 3029–3056) is shown in purple and the ka9b (aa 2942–2956) and ka10 (aa 3002–3026) is shown in pink. The αC-helix (aa 2722–2741), which is a part of the N-lobe and contains

*Figure 1 continued on next page*

Figure 1 continued

residues critical for catalysis is shown in blue. ATP and residues critical for catalysis are shown in the stick model. $Mg^{2+}$ ions are shown as spheres. The black dotted loops indicate the disordered region between ka9b and ka10. The amino acids that are conserved between human ATM and mTOR and mutated in TCGA cases are shown in red.

The following figure supplement is available for figure 1:

**Figure supplement 1.** Cancer associated ATM mutations are enriched for kinase domain missense mutations that disrupt kinase activity.

## Expression of Atm-KD is more oncogenic than loss of Atm

To determine the oncogenic properties of ATM-KD protein and circumvent the embryonic lethality of the $Atm^{KD/-}$ mice (*Yamamoto et al., 2012*), we generated littermate matched $VavCre^+Atm^{C/KD}$ (VKD) and $VavCre^+Atm^{C/C}$ (VN, N for null) mice. In these mice, the HSC-specific Cre recombinase (*VavCre*) (*de Boer et al., 2003*) efficiently inactivates the Atm conditional ($Atm^C$) allele (*Zha et al., 2008*; *Callén et al., 2009*) in all blood cells including lymphocytes (*Figure 2—figure supplement 1A*). VKD and VN mice display defects in B cell CSR and T cell development similar to $Atm^{-/-}$ mice, namely 50% decrease of CSR (to IgG1), reduced expression of surface-TCRβ/CD3 in $CD4^+CD8^+$ double-positive (DP) cells, and partial blockage at DP to $CD4^+CD8^-$ or $CD4^-CD8^+$ single-positive (SP) T cell transitions (*Borghesani et al., 2000*; *Lumsden et al., 2004*; *Reina-San-Martin et al., 2004*) (*Figure 2—figure supplement 1B–C*), supporting efficient inactivation of $Atm^C$ allele in VKD and VN lymphocytes. Immature T cells with productive TCRβ rearrangements undergo rapid proliferation and expansion in the $CD4^-CD8^-$ double-negative (DN) 3 stage ($CD44^-CD25^+$). VKD thymocytes, but not VN or $Atm^{-/-}$ thymocytes, are partially blocked at the DN3 stage, consistent with proliferation defects in $Atm^{KD/-}$ cell (*Yamamoto et al., 2012*) (*Figure 2A–B*). As a result, thymocyte number in VKD mice was reduced ~60% compared to their VN littermates (*Figure 2—figure supplement 1C*).

Despite low thymocyte counts, 75% of VKD mice succumbed to lymphomas, representing a 34% increase over the 56% life-time risk among VN mice (*Figure 2D*). Furthermore, the median survival of thymic lymphoma bearing VKD mice is ~35 days shorter than that of VN mice (104 and 139 days respectively, p=0.03) (*Figure 2E*). The thymic lymphomas from the VN mice and VKD mice are both clonal, immature (TCRβ/CD3^low) (*Figure 2—figure supplement 1D–E*) with frequent trisomy 15, have focal amplifications around the TCRα/δ loci, and have hemizygous deletion of the telomeric portion of chromosome 12, as is the case for previously characterized $Atm^{-/-}$ thymic lymphomas (*Jiang et al., 2015b*; *Zha et al., 2010*) (*Figure 2—figure supplement 2A*). About 8% of the VKD, but none of the VN mice, also developed clonal pro-B cell lymphomas ($B220^+CD43^+IgM^-$) (*Figure 2—figure supplement 2B–C*). Comparative genome hybridization (CGH) showed that all VKD thymic lymphomas, but none of the VN controls, carried deep deletions at the *Pten* locus (*Figure 2F* and *Figure 2—figure supplement 2A*). Immunoblotting of additional lymphomas revealed severe reductions of Pten protein levels in all but one of the VKD tumors, while most VN tumors retained Pten expression. The only Pten-expressing VKD tumor (6684) appeared to have lost Atm-KD expression (*Figure 2G*). *Pten* deletion was previously found in ~25% of germline $Atm^{-/-}$ or $Tp53^{-/-}$ thymic lymphomas and was thought to occur in the early progenitors before T cell commitment (*Zha et al., 2010*; *Dudgeon et al., 2014*). Based on our findings, we propose that expression of Atm-KD protein further increases genomic instability in lymphoid progenitors/HSCs, eventually led to frequent *Pten* deletion and early lymphomas in the VKD mice. Consistent with this model, frequency of chromatid breaks are significantly higher in VKD T cells, than in VN or $Atm^{+/+}$ T cells (*Figure 2H*).

## $Atm^{KD/-}$ cells and $Atm^{KD/-}$ leukemia are hypersensitive to Topoisomerase I inhibitors

We hypothesized that the additional DNA repair defects in $Atm^{KD/-}$ cells would confer hypersensitivity to certain genotoxic chemotherapy that could be used to target ATM-mutated cancers. To test this, we derived $Rosa^{+/CreERT2}Atm^{+/C}$, $Rosa^{+/CreERT2}Atm^{C/-}$ and $Rosa^{+/CreERT2}Atm^{C/KD}$ murine embryonic fibroblasts (MEFs), in which 4-Hydroxytamoxifen (4-OHT, 200 nM) induces nuclear translocation of ER-fused Cre-recombinase and effectively inactivates the $Atm^C$ allele to generate $Atm^{+/-}$, $Atm^{-/-}$ and $Atm^{KD/-}$ MEFs (*Figure 3—figure supplement 1A*) (*Yamamoto et al., 2012*). All the experiments were repeated ≥3 times in several independently derived and freshly deleted cells to avoid

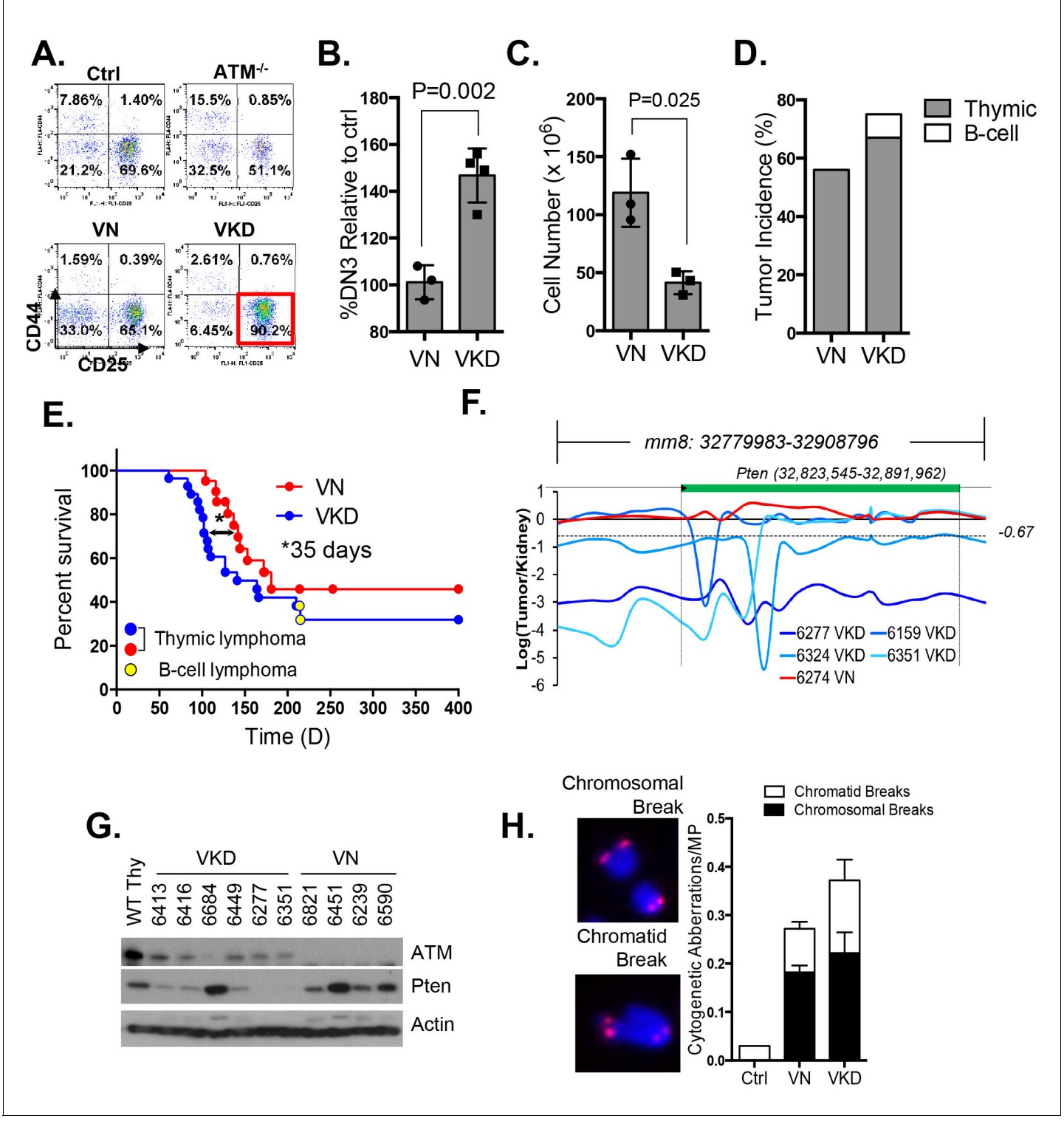

**Figure 2.** Expression of ATM-KD protein (Atm$^{KD/-}$) is more oncogenic than loss of ATM (Atm$^{-/-}$). (**A**) Representative FACS analyses of DN thymocytes from 4-week old *VavCre-Atm*$^{C/KD}$ (Ctrl, littermate), germline Atm-knockout (*Atm*$^{-/-}$), *VavCre*$^+$*Atm*$^{C/C}$ (VN) and *VavCre*$^+$*Atm*$^{C/KD}$ (VKD) mice. DN cells are defined as thymocytes negative for surface staining of CD8, CD4 and TCRγδ. (**B**) The average relative DN3% (CD44$^-$CD25$^+$) (among all DN thymocytes) in VN (n = 3) and VKD (n = 4) mice. The relative DN3% was calculated by normalizing to the absolute DN3% to the DN3% of control littermate *VavCre*$^-$ (cre negative) mice stained at the same time. The error bars represent SEMs. (**C**) The averages and standard SEMs of total thymocyte number from 4–8 weeks old VN (n = 3) and VKD (n = 3) mice. (**D**) The life time risk of B or T cell lymphomas in VN, VKD mice in 400 days. (**E**) Kaplan-Meier (K-M) survival curve of littermate matched VN (n = 18) and VKD (n = 24) mice. The red and blue dots denote thymic lymphomas from the VN and VKD cohorts,

*Figure 2 continued on next page*

*Figure 2 continued*

respectively. The yellow dots denote the B cell lymphomas in the VKD cohort. The $T_{1/2}$ for thymic lymphoma is 139.5 days (VN) and 104.0 days (VKD). The asterisk marks the difference in $T_{1/2}$ of thymic lymphoma development between the two cohorts. *p=0.03 per Mantel-Cox/log-rank test. (F) Copy number analyses of the region around *Pten* (chromosome 19, *mm8: 32,779,983–32,908,796*). The y-axis is the natural log ratio of tumor/kidney genomic DNA from the same mouse. Log ratio of −0.67 (dotted line) indicates heterozygous deletion. (G) Immunoblot of Pten, total Atm and β-actin for $Atm^{+/+}$ (WT Thy), VKD, and VN thymic lymphomas. (H) Telomere FISH analyses of metaphases spreads from ConA activated T-cells (72 hr). Bar graphs represent the average and SEMs of the number of cytogenetic aberrations per metaphase (MP). All p-values in this figure were calculated using a two-tailed student's t-test assuming unequal variances, unless otherwise noted.

The following figure supplements are available for figure 2:

**Figure supplement 1.** Lymphocyte development in VKD and VN mice.

**Figure supplement 2.** Analyses of the T cell lymphomas from VKD and VN mice.

secondary alterations in genomically unstable Atm-deficient cells. While both $Atm^{KD/-}$ and $Atm^{-/-}$ MEFs were similarly sensitive to ionizing radiation (IR), $Atm^{KD/-}$ cells were significantly more sensitive to the Topo1 inhibitor camptothecin (CPT) than $Atm^{-/-}$ and $Atm^{+/-}$ cells (*Figure 3A–B*). The CPT hypersensitivity of $Atm^{KD/-}$ cells does not appear to reflect an inability to process blocked DNA ends or overcome replication blockage in general, as $Atm^{KD/-}$ cells were not hypersensitive to etoposide, a Topoisomerase *II* inhibitor, nor to two replication blocking agents, hydroxyurea (HU) or aphidicolin (APH), (*Figure 3C* and *Figure 3—figure supplement 1B*). Moreover, $Atm^{KD/-}$ primary T cells and the human cancer cell lines that express ATM-KD protein identified in *Figure 1—figure supplement 1F* are also hypersensitive to CPT compared to corresponding $ATM^{+/+}$ or $Atm^{-/-}$ controls (*Figure 3D* and *Figure 3—figure supplement 1E*). Finally, shRNA knockdown of mutated ATM in Granta519 human lymphomas cell lines, moderately, yet significantly, rescued the CPT hypersensitivity and CPT induced apoptosis associated with the expression of catalytically inactive ATM (*Figure 3E* and *Figure 3—figure supplement 1C and D*).

In animal models, irinotecan, a clinical Topo1 inhibitor, eradicated activated-Notch1 (Notch1-ΔE-IRES-GFP) driven $Atm^{KD/-}$ T cell leukemia, but not the isogenic parental $Atm^{KD/C}$ leukemia in vivo, determined by the absence of GFP$^+$ leukemia cells in the spleen, reduced spleen size and weight, lack of infiltrated leukemia blasts and increased TUNEL$^+$ apoptotic cells (*Figure 4A–F* and *Figure 4—figure supplement 1A–B*). When similar experiments were performed on isogenic $Atm^{C/C}$ and corresponding $Atm^{-/-}$ leukemia, irinotecan has no significant benefit in either groups (*Figure 4—figure supplement 1C–F*, p≥0.05 in all pairs), suggesting loss of ATM-mediated DNA damage responses alone is not sufficient to explain the hypersensitivity of $Atm^{KD/-}$ leukemia to Topo1 inhibitors. Together these findings identified the expression of ATM-KD as a biomarker for Topo1 inhibitors sensitivity.

## ATM-KD blocks replication-dependent removal of Topo-isomerase I DNA adducts at the step of strand cleavage

Topo1 inhibitors prevent DNA-religation by Topo1, and trap Topo1 in a covalent DNA complex (Top1cc) at the 3'-end of single-strand DNA (ssDNA) nicks generated by Topo1 cleavage (*Pommier, 2006*). If not removed, Top1cc interferes with transcription and replication to elicit cytotoxic effects. We found that $Atm^{KD/-}$, but not $Atm^{-/-}$ or $Atm^{+/+}$ MEFs accumulated Top1cc even at low CPT concentrations (0.1 μM) (*Figure 5A* and *Figure 5—figure supplement 1A and B*). ATM kinase inhibitor (ATMi, KU55933, 15 μM) also leads to CPT-dependent Top1cc-accumulation in $Atm^{+/-}$ MEFs (*Figure 5A*). At high CPT concentration (15 μM), the Top1cc level is greatest in $Atm^{KD/-}$ cells, followed by $Atm^{-/-}$, then $Atm^{+/+}$ (*Figure 5A* and *Figure 5—figure supplement 1A and B*). Meanwhile, CPT-induced degradation of full-length Topo1, necessary for its subsequent removal (*Pommier, 2006*), is not significantly affected in $Atm^{KD/-}$ or $Atm^{-/-}$ MEFs (*Figure 5B*). Tyrosyl-DNA phosphodiesterase 1 (TDP1) or 3' flapase (*e.g.* XPF) can remove Top1cc independent of DNA replication (*Pommier, 2006*) (*Figure 5C*). In quiescent neurons, ATM promotes transcription-dependent removal of Top1cc potentially by facilitating Topo1 degradation independent of its kinase activity or MRN (*Alagoz et al., 2013*; *Katyal et al., 2014*). While this function of ATM might

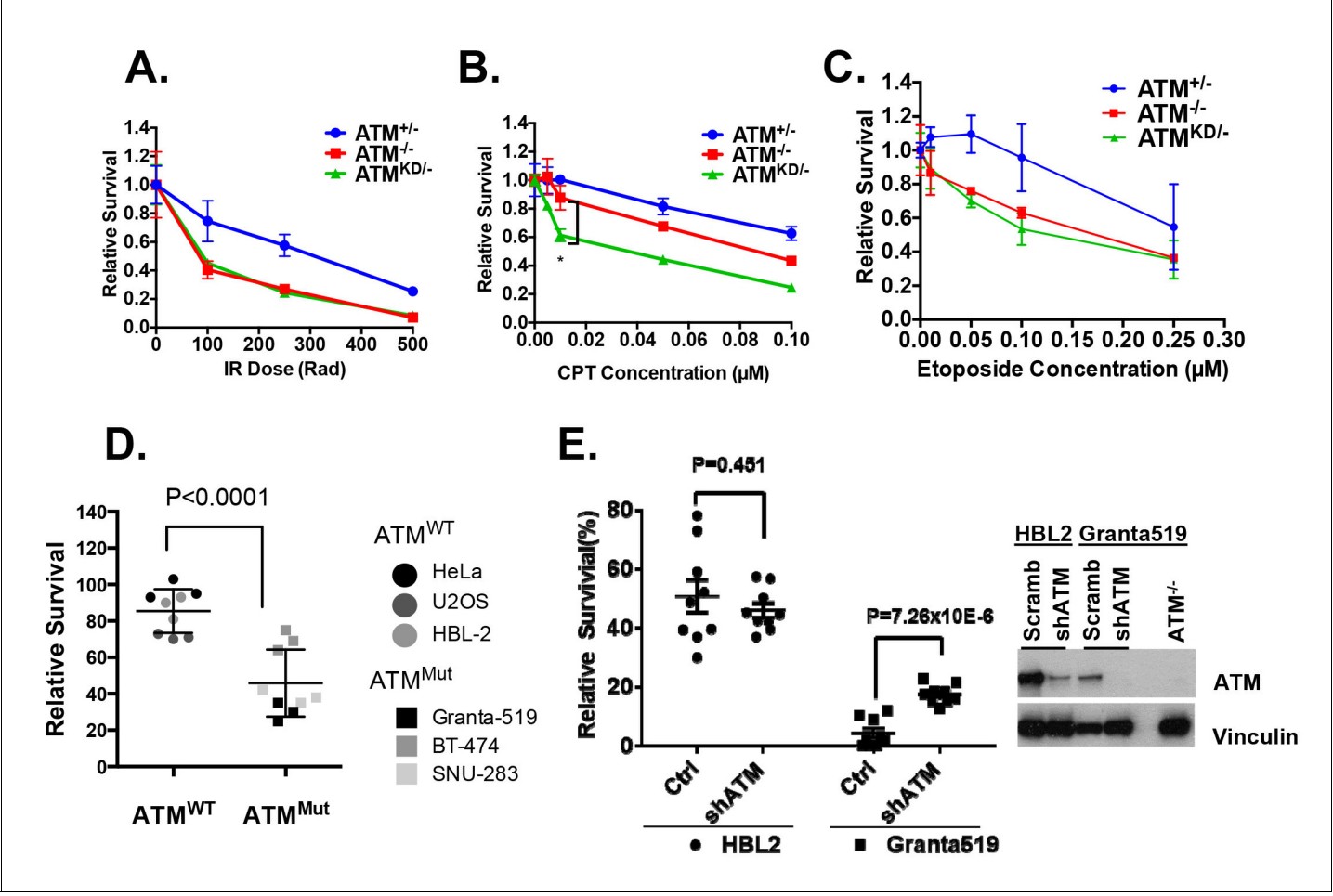

**Figure 3.** *Atm*^KD/- murine embryonic fibroblasts (MEFs) and human cancer cell lines are hypersensitive to Topo isomerase I inhibitors in vitro. Representative sensitivity plots of *Atm*^+/-, *Atm*^-/- and *Atm*^KD/- MEFs to (**A**) IR, (**B**) Camptothecin (CPT) and (**C**) Etoposide. *p<0.001. At least three independent experiments on two independently derived MEF lines from each genotype were performed. (**D**) Sensitivity of 6 human tumor cell lines with wildtype (*ATM*^+/+) or mutant ATM (*ATM*^Mut/- or Mut/Mut) to 2.5 nM CPT observed after 48 hr of treatment. (**E**) shRNA mediated knockdown of ATM in Granta591 human lymphoma cell lines rescued the CPT sensitivity in vitro (after 72 hr of 5 nM CPT treatment). Three independent experiments were performed. The p value were obtained based on student t-test.

The following figure supplement is available for figure 3:

**Figure supplement 1.** Analyses of immortalized *Atm*^KD/- and control MEFs.

contribute to the moderate accumulation of Top1cc in CPT treated *Atm*^-/- MEFs at high CPT concentrations (*Figure 5A*), the kinase-dependent and protein-dependent role of ATM in Top1cc removal revealed in proliferating *Atm*^KD/- MEFs appears to be independent of Topo1 degradation (*Figure 5B*). In this context, a recent study using Xenopus extract suggests that collision of the replication forks with protein conjugated to DNA would lead to rapid and robust proteolytic degradation of the conjugated proteins, which might mask the moderate effects of ATM in Top1cc-proteolytic degradation described in none proliferative neuronal cells (*Duxin et al., 2014*). During replication, it is proposed that the collision of the DNA replication forks with Top1cc blocks the replication on the involved strand, leads to uncoupling of the leading strand and lagging strand syntheses and fork reversal. Eventually a subset of the stalled replication forks are cleaved to remove Top1cc (as diagrammed in *Figure 5C*). The resulting single-ended DSBs generated from fork cleavage can only be repaired by HR, not NHEJ. Resolution of the holiday junction intermediates during HR generates crossover (CO), thus CPT is known to induce SCEs (*Figure 5C*). CPT (1 µM) induced SCEs were reduced in *Atm*^KD/- MEFs, but not *Atm*^-/- MEFs, and ATM inhibitor decreased CPT-induced SCEs in

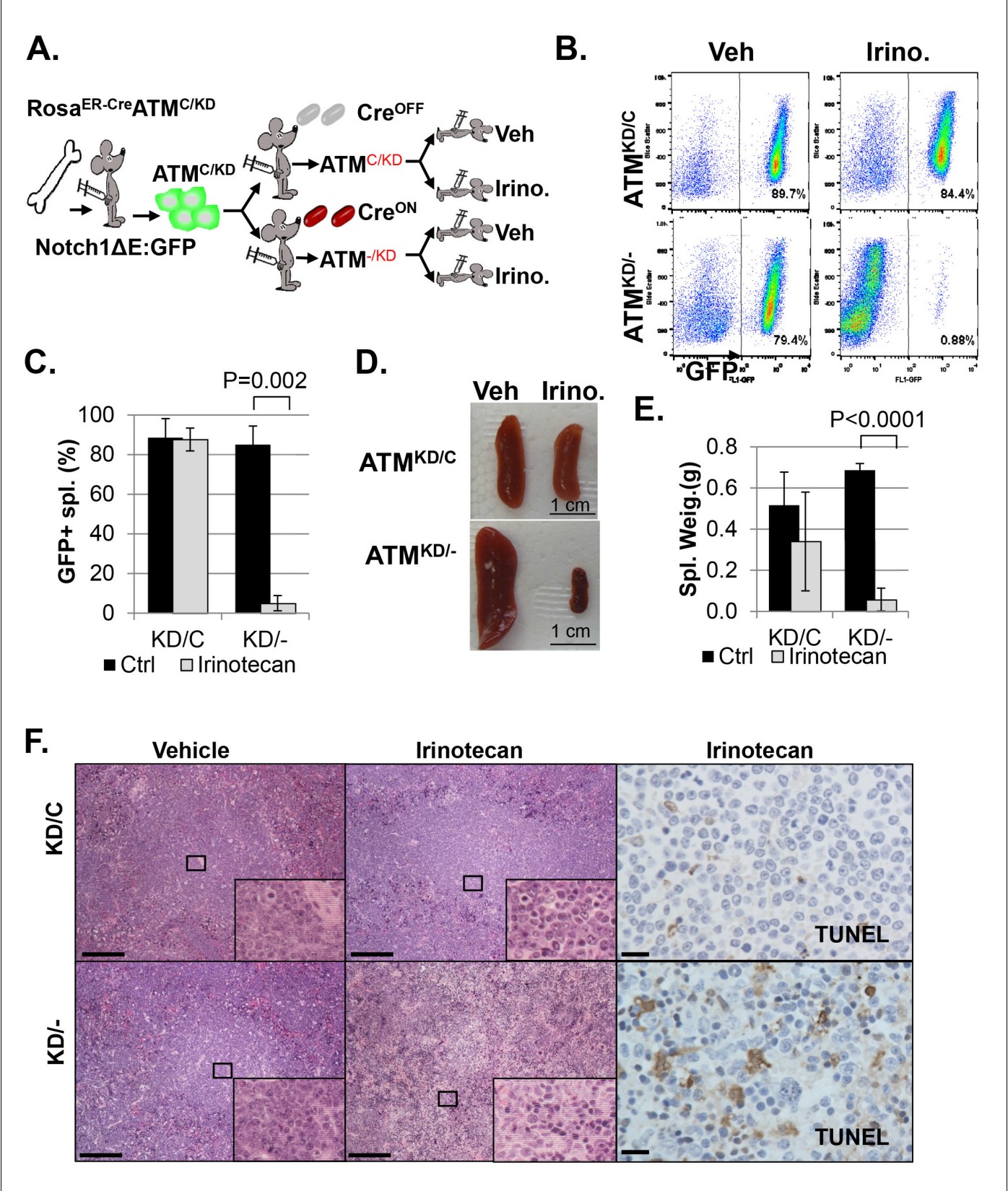

**Figure 4.** *Atm*[KD/-] leukemia are hypersensitive to Topo isomerase I in vivo. (**A**) Experimental scheme outlining the treatment of isogenic *Atm*[KD/-] and *Atm*[KD/C] leukemia in vivo. Please see on-line methods for details. Clonal Rosa[+/CreERT2] Atm[C/KD] leukemia was established by transducing the bone

*Figure 4 continued on next page*

*Figure 4 continued*

marrow with activated Notch and GFP. Isogenic $Atm^{KD/-}$ or $Atm^{KD/C}$ leukemia were then established and confirmed in secondary recipients upon exposure to oral tamoxifen, which induces Cre nuclear translocation. $1 \times 10^6$ $Atm^{KD/-}$ and $Atm^{KD/C}$ leukemic blasts were injected into tertiary recipients, which were treated with daily doses of Irinotecan (10 mg/kg) or vehicle for 5 consecutive days. Representative FACS analyses for GFP$^+$ leukemia blast (B) and pictures (D) of spleens from mice transduced with isogenic $Atm^{KD/C}$ or $Atm^{KD/-}$ leukemia and treated or not treated with irinotecan (Irino, 10 mg/Kg for 5 days). The untreated mice received vehicle (Veh). Quantification (Average ± SEMs) of GFP$^+$% splenocytes (C) and spleen weights (E) of mice transduced with $Atm^{KD/C}$ or $Atm^{KD/-}$ leukemia and treated with either vehicle (black) or Irinotecan (grey) for 5 days. (F) Representative field of H&E (left 2 panels) and TUNEL-stained spleen sections Scale Bars: 200 µm (in H&E section), 20 µm (TUNEL).

The following figure supplement is available for figure 4:

**Figure supplement 1.** Analyses of isogenic paired $Atm^{KD/-}$ vs $Atm^{KD/C}$ or $Atm^{-/-}$ vs $Atm^{C/C}$ leukemia.

$Atm^{+/-}$, but not in $Atm^{-/-}$ MEFs (*Figure 5D*), suggesting a defect at replication-dependent removal of Top1cc in the $Atm^{KD/-}$ cells. ATM kinase inhibitor did not suppress SCEs in Blmhelicase-deficient ($Blm^{-/-}$) MEFs, suggesting that the lack of SCEs could not be simply explained by the bias in CO-generating holiday junction resolvases (*e.g.* MUS81, SLX1/4, GEN1) (*Figure 5—figure supplement 1C*) (*Chan and West, 2014*).

In this context, an integrated (at the Pim1 locus) DR-GFP reporter (*Moynahan et al., 1999*) revealed a 50% reduction of HR in $Atm^{KD/-}$ embryonic stem (ES) cells, but not in $Atm^{-/-}$ cells. ATM inhibitor reduced HR in $Atm^{+/+}$, but not $Atm^{-/-}$ ES cells (*Figure 4D*) (*Rass et al., 2013*; *Kass et al., 2013*), consistent with potential HR defects. End resection that reveals single strand DNA (ssDNA) and loading of Rad51 protein that is necessary for homology search are two early events required for HR. Yet *IR* induced phosphorylated-RPA (T21) and Rad51 foci, were not significantly compromised in $Atm^{KD/-}$ cells (*Figure 6A* and *Figure 6—figure supplement 1A*) (*Shakya et al., 2011*). These findings imply that Atm-KD does NOT block RAD51 loading to resected DSBs and suggest that either a later step of HR is affected in $Atm^{KD/-}$ cells or other mechanisms exist to explain the lack of CPT-induced SCEs in $Atm^{KD/-}$ cells. In this regard, we found that *CPT*-induced Rad51 foci were markedly decreased in $Atm^{KD/-}$ MEFs (*Figure 6B*). Moreover CPT-induced DSBs measured by phosphorylated H2AX (γ-H2AX) and the neutral comet assay were also attenuated in $Atm^{KD/-}$ cells (but not $Atm^{-/-}$ cells) and by ATM kinase inhibitor (*Figure 6C–E*). In the same cells, CPT effectively increased alkaline comet tails, an indicator of both single and double stand breaks, in $Atm^{+/-}$, $Atm^{-/-}$ and $Atm^{KD/-}$ cells (*Figure 6F*). Together these findings indicate that ATM-KD protein suppressed CPT-induced DSBs formation during replication and thereby reduced SCEs.

Upon DNA DSBs, Mre11, along with the MRN complex recruits ATM to the DSBs and activates ATM kinase activity. Notably, Mre11 also binds to ssDNA and the affinity of Mre11 to ssDNA is higher than to dsDNA and is independent of DNA ends (*de Jager et al., 2001*; *Usui et al., 1998*; *Paull and Gellert, 1999*). Extended ssDNA (potentially without ends) is likely the structure that accumulated at stalled and uncoupled replication forks (*Figure 5C*). Correspondingly, recent iPOND experiments have identified MRE11 at the stalled replication fork in vivo (*Sirbu et al., 2011*). In this context, we found that ATM kinase inhibitor prevents CPT induced DSBs formation in $Mre11^{+/-}$ but not in $Mre11^{-/-}$ MEFs. The presence of a nuclease deficient Mre11 ($Mre11^{nuc/-}$) that binds and recruits ATM (*Buis et al., 2008*) is sufficient to abolished CPT induced DSBs in the presence of ATM kinase inhibitor (*Figure 7A*, *Figure 7—figure supplement 1A–C*). Together these findings support a model, in which ATM-KD protein physically blocks CPT induced DSBs formation at replication forks upon recruitment by MRN.

Collision with Top1cc uncouples the progression of the leading and lagging strands, and causes accumulation of ssDNA (*Figure 5C*), which were thought to be the precursor of 'fork reversal' that converts a replication fork into a four-way junction (*Figure 4—figure supplement 1A*) (*Zellweger et al., 2015*; *Berti et al., 2013*). Fork reversal slows replication. In one hand, fork reversal could allow the replication machinery to bypass the lesion using the newly synthesized stand as the template. On the other hand, the four-way junctions, if they persist, could be recognized by structural specific nucleases (*e.g.* MUS81, SLX1/4) implicated in CPT induced DSBs (*Regairaz et al., 2011*). In this regard, we reported that CPT induced RPA phosphorylation at Thr 21, a marker for ssDNA and ssDNA induced CHK1 phosphorylation by ATR are normal in $Atm^{KD/-}$ cells and ATM

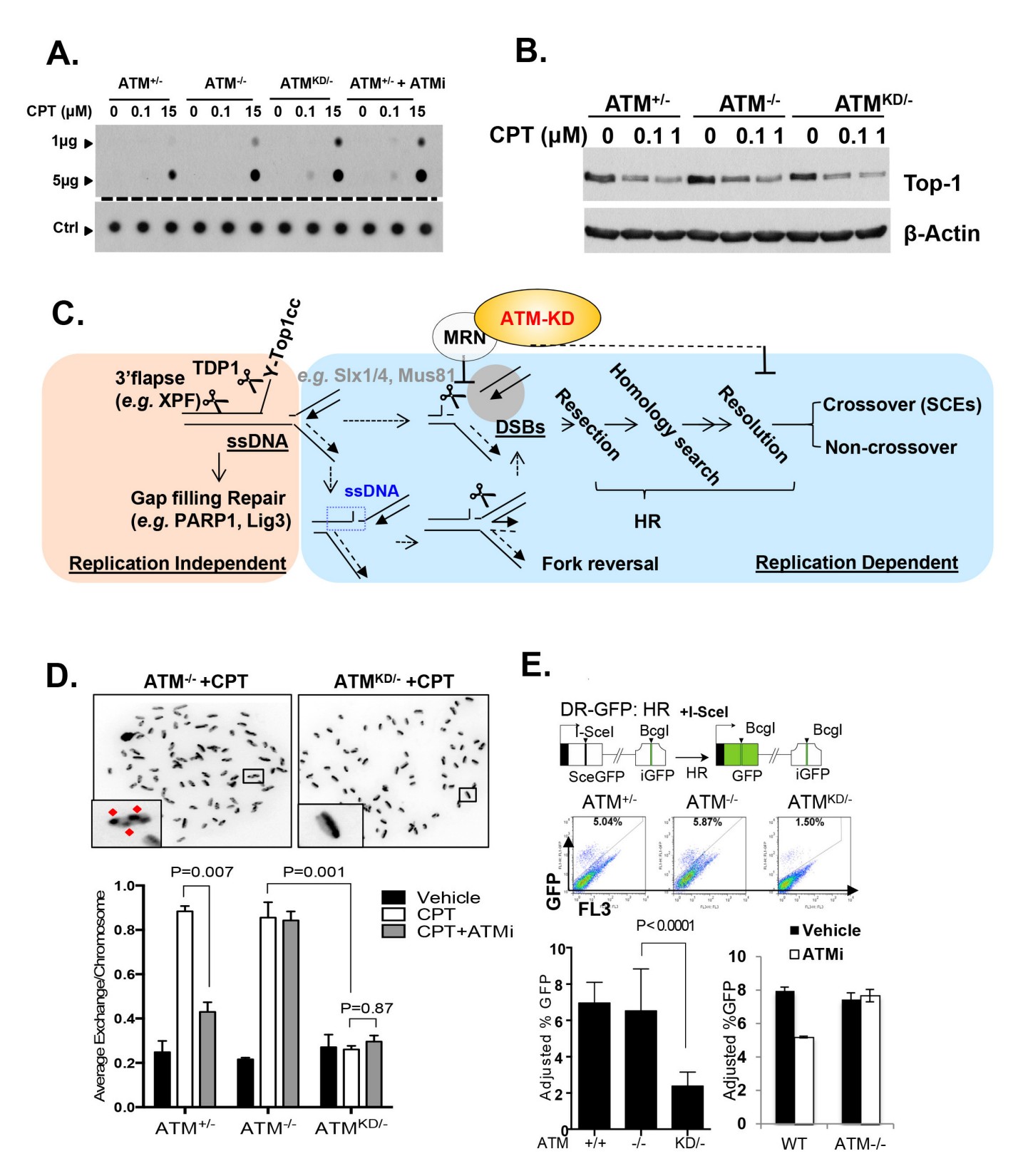

**Figure 5.** Accumulation of Top1cc and reduced homologous recombination in *Atm*[KD/-] cells. (**A**) Representative In-vitro Complex of Enzymes (ICE) assay (see Materials and method). The top panel was blotted by anti-Topo1 antibody and the bottom panel was probed by p32 labeled total mouse

*Figure 5 continued on next page*

*Figure 5 continued*

genomic DNA. The amounts of genomic DNA (in µg) loaded on each row are marked on the left. (**B**) Western blot for full length Topo1 on MEFs treated with indicated concentrations of CPT for 2 hr. (**C**) Diagram of replication-dependent (blue shade) and replication-independent (pink shade) Top1cc removal pathways. Top1cc is removed by TDP1 or 3' flapases to generate a single stand gap, which can be repaired by PARP1 and Ligase3-mediated pathways *independent* of replication. In replicating cells, replication forks collide with Top1cc, lead to uncoupling of the leading and the lagging strand, accumulation of single strand DNA (blue dashed box), which promotes fork reversal and eventually a subset of the forks were cleaved. The stalled fork could also be directly converted to breaks (top row, dash arrow). The resulting single-ended DSBs (grey shade circle) can only be repaired by HR (through end resection, homology search and resolution) to generate cross over (CO) (scored as SCEs) or non–crossover. ATM-KD protein selectively inhibits fork breakage in a MRN dependent manner, and could also potentially suppress the same nucleases (Mus81/SLX4) implicated in holiday junction resolution in a later step of HR. (**D**) Representative images and the average (and SEMs) of SCEs per chromosome for vehicle (DMSO) or CPT (1 µM) treated MEFs. Red diamonds point to exchange events. (**E**) Diagram of the DR-GFP reporter, representative FACS plots of the GFP+ cells 48 hr after I-SceI transfection. The bar graphs represent the adjusted GFP%, calculated based on raw GFP% and the I-SceI transfection efficiency. At least two independent ES cell lines per genotype were assayed in >3 independent experiments. Bar graphs represent the average ± SEMs. In the right panel, the cells were either treated with vehicle (DMSO) or ATMi (15 µM Ku55933) for 36 hr (12 hr after I-SceI transfection).

The following figure supplement is available for figure 5:

**Figure supplement 1.** Analyses of immortalized *Atm*[KD/-] MEFs and Mre11-deficient cells.

kinase inhibitor-treated Atm[+/-] cells (***Figure 6D***, ***Figure 6—figure supplement 1B***, ***Figure 7—figure supplement 1E***). Moreover, we found that ATM kinase inhibitor did not alter the frequency or size of CPT induced ssDNA, or the frequency of actual fork reversal in *Atm*[+/-] cells measured by electron-microscopy (***Zellweger et al., 2015***)(***Figure 7B and C***). Correspondingly, CPT significantly reduced fork velocity in *Atm*[KD/-] as well as control *Atm*[+/+] or *Atm*[-/-] cells (***Figure 7D***), consistent with fork reversal. We noted that baseline fork progression is slowest in *Atm*[KD/-] cells (***Figure 7D***), potentially reflecting defects in resolving spontaneous Top1cc or related lesions, such as Topo1-processed ribo-nucleotide mis-incorporation (***Kim et al., 2011***). Together, our findings support a model, in which ATM is recruited to stalled replication fork by MRE11 (or MRN complex), where ATM kinase activity is necessary to release ATM and allow fork cleavage without affecting fork uncoupling and fork reversal. This role of ATM at the replication fork explains the increased genomic instability, especially chromatid breaks, and hypersensitivity to CPT in murine cells as well as human cancer cells that express catalytically inactive ATM protein.

## Discussion

*ATM* is a tumor suppressor gene that is frequently inactivated in human cancers. While previous studies have mostly focused on the complete loss of ATM (*Atm*[-/-], truncating mutations), recent sequencing analyses identified a large number of missense ATM mutations in human cancers with limited information on their biological function. Here we report that cancer-associated ATM mis-sense mutations are highly enriched in the kinase domain. Using murine models, we showed that expression of ATM-KD causes cancer more frequently and rapidly than loss of ATM, and the *Atm*[KD/-] cancers are selectively hypersensitive to Topo1 inhibitors. As such, Topo1 inhibitors could potentially be a *targeted* therapy for tumors with ATM kinase domain mutations. Mechanistically, we found that ATM-KD physically blocks CPT-induced DSBs formation during replication in a MRN-dependent manner. Given that both *Atm*[KD/-] cells and ATM kinase inhibitor treated *Atm*[+/-] cells display defects in replication-dependent Top1cc removal, our data revealed not only a neomorphic function associ-ated with kinase domain point mutations of ATM, but also a previously unappreciated auto-phos-phorylation dependent structural function of normal ATM that is not apparent in *Atm*[-/-] cells (due to the lack of both ATM protein and ATM kinase activity). This new function of ATM expands the reper-toire through which ATM functions as a tumor suppressor and has implications on DNA repair and cancer therapy.

Despite the ~two fold increased breast cancer risk in heterozygous ATM mutation carriers (***Renwick et al., 2006***; ***Ahmed and Rahman, 2006***) and 1–2% carrier rate in Caucasians, truncating ATM mutations are exceedingly rare (<1/400, in contrast to the predicted 8%) in early-onset breast cancers patients (***FitzGerald et al., 1997***). This and other findings (***Spring et al., 2002***)

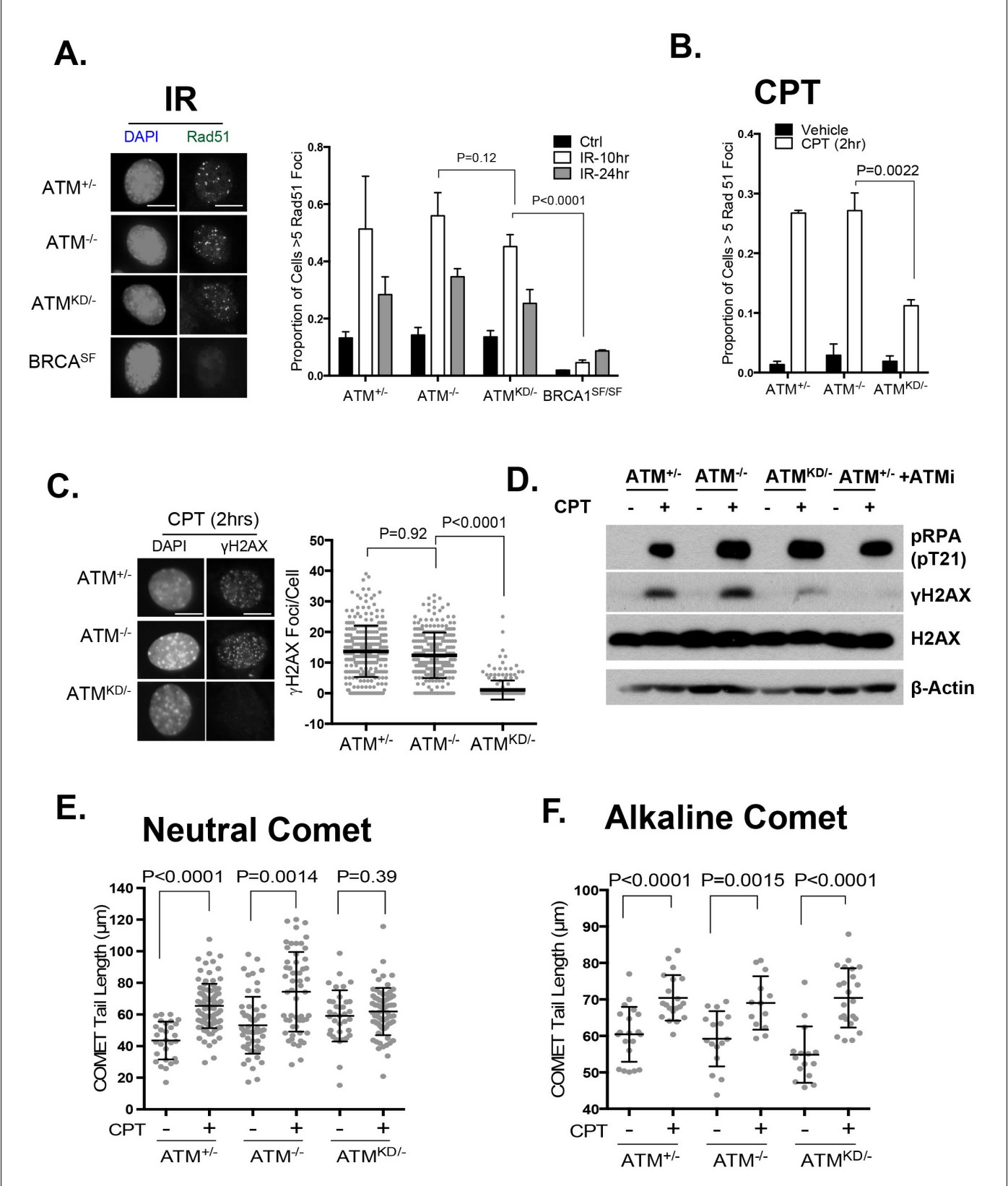

**Figure 6.** ATM-KD blocks CPT-induced double stand breaks formation in replicating cells. (**A**) Representative images and the frequency of cells with >5 Rad51 foci per cell (Average ± SEMs) at 10 hr after 5 Gy of *IR*. Brca1^SF/SF^ cells homozygous for the S1598F mutation(***Shakya et al., 2011***) were used as a
*Figure 6 continued on next page*

*Figure 6 continued*

control for the lack of Rad51 foci. Scale bar ~10 µm. (B) The proportion of cells with >5 Rad51 foci after *CPT* (0.1 µM, 2 hr) treatment. The bar graph represents the average and SEMs from 3 independent experiments. All p-values in this figure were calculated based on two-tailed student's t-test assuming unequal variances. (C) Representative images and quantification of CPT-induced γH2AX foci (0.1 µM, 2 hr). The dot plot represents average and SEMs in three biological replicates with >200 nuclei per genotype per condition. Scale bar ~10 µm. (D) Western blots for pRPA(T21), γH2AX, and total H2AX of cells treated with either Vehicle (−, DMSO) or CPT (+, 0.1 µM) for 2 hr. ATM inhibitor (15 µM, Ku55933) was added 1 hr prior to CPT/ vehicle treatment as indicated. (E) Quantification of Neutral COMET tail lengths in cells treated with Vehicle (−, DMSO) or CPT (+, 0.1 µM) for 1 hr. The graph represents the average ± SEMs from two independent experiments with over 50 comets quantified for each. (F) Quantification of Alkaline COMET tail lengths of cells treated with Vehicle (−, DMSO) or CPT (+, 0.1 µM) for 1 hr. Two replicates of the experiments were performed with over 50 comets quantified for each.

The following figure supplement is available for figure 6:

**Figure supplement 1.** Analyses of cell cycle dependent reduction of CPT induced DSBs.

led to the speculation that ATM mutations are different in cancer compared to A-T (*Gatti et al., 1999*). Here we reported that, while $Atm^{+(C)/KD}$ mice are normal (*Yamamoto et al., 2012*), somatic inactivation of the $Atm^C$ allele in VKD mice led to aggressive lymphomas. As such, while missense ATM mutations in the kinase domain are not compatible with embryonic development when homozygous and are not found in A-T, they could predispose carriers to cancers, and maybe have more potent oncogenic potential than A-T causing truncating mutations. Notably, loss of ATM preferentially affects NHEJ (*Franco et al., 2006*) and predisposes patients primarily to lymphomas. Here we show that $Atm^{KD}$ also compromises HR, a pathway implicated in breast, ovarian, pancreatic and prostate cancers, in which recurrent ATM missense mutations were recently identified (*Roberts et al., 2012*; *Cremona and Behrens, 2014*). MRE11 binds to ssDNA independent of DNA ends in vitro (*de Jager et al., 2001*; *Usui et al., 1998*; *Paull and Gellert, 1999*) and MRE11 has been identified at stalled replication forks in vivo (*Sirbu et al., 2011*). Our results also revealed that the ATM-KD protein physically blocks strand cleavage in a MRN-dependent manner, further suggesting the level of mutant ATM protein and MRN status can modify both the cancer risk and therapeutic responses. ATM kinase inhibitor seems to reduce CPT-induced DSBs more efficiently in $Mre11^{Nuc/-}$ cells, than in $Mre^{+/-}$ cells (*Figure 7A*, *Figure 7—figure supplement 1A–C*). This result suggests that Mre11 nuclease activity might contribute to the removal of ATM or the termination of ATM activation. Further investigation is likely needed to clarify this issue. We noted that loss of sae2, a modulator of Mre11 nuclease activity in yeast, causes persistent DNA damage responses mediated by *tel1/mec1*, yeast orthologs of *ATM* and *ATR*. Two recent studies have identified mre11 mutations that rescued the hyper activation of mec1 in sae2 deficient yeast. (*Chen et al., 2015*; *Puddu et al., 2015*). Mre11 nuclease might also contribute to fork cleavage directly. In this context, we noted that Mre11 nuclease inhibitors – either PFM01 or Mirin (*Shibata et al., 2014*), reduced CPT induced γ-H2AX in both $Atm^{+/-}$ and $Atm^{-/-}$ cells, supporting an ATM independent function of Mre11 in fork cleavage (*Figure 7—figure supplement 1D*). However, given the toxicity associated with Mre11 nuclease inhibitors, further experiments are warranted to understand the molecular details of MRE11 function in fork cleavage. Together, these findings support our model in which MRE11, together with the MRN complex recruits ATM to the stalled replication forks, while the kinase activity of ATM is important for fork cleavage in the presence of ATM protein.

The mechanism identified here also suggests that $Atm^{KD/-}$ tumors should be hyper-sensitive to agents targeting genes involved in Top1cc removal/repair (*e.g.* PARPs,TDP1) (*Pommier, 2009*; *Murai et al., 2014*) andgenotoxic agents that require similar 'strand-cleavage' mechanisms for repair (*e.g.* crosslink agents –like Mitomycin C) (*Zhang and Walter, 2014*). Consistent with these predictions, $Atm^{KD/-}$ T cells are hypersensitive to PARP inhibitor alone or in combination with CPT (*Figure 3—figure supplement 1E*) and $Atm^{KD/-}$, but not $Atm^{-/-}$ MEFs, are hypersensitive to Mitomycin C (*Figure 7—figure supplement 1F*). Together, these findings further expand the therapeutic options for ATM mutated cancers. The similarity between $Atm^{KD/-}$ cells and ATM kinase inhibitor treated $Atm^{+/-(+)}$ cells further suggest that ATM inhibitors, which have recently entered clinical trials, could potentially increase the efficacy of Topo1 inhibitors in $ATM^{+/+}$ cancers. Finally, the lack of CPT-

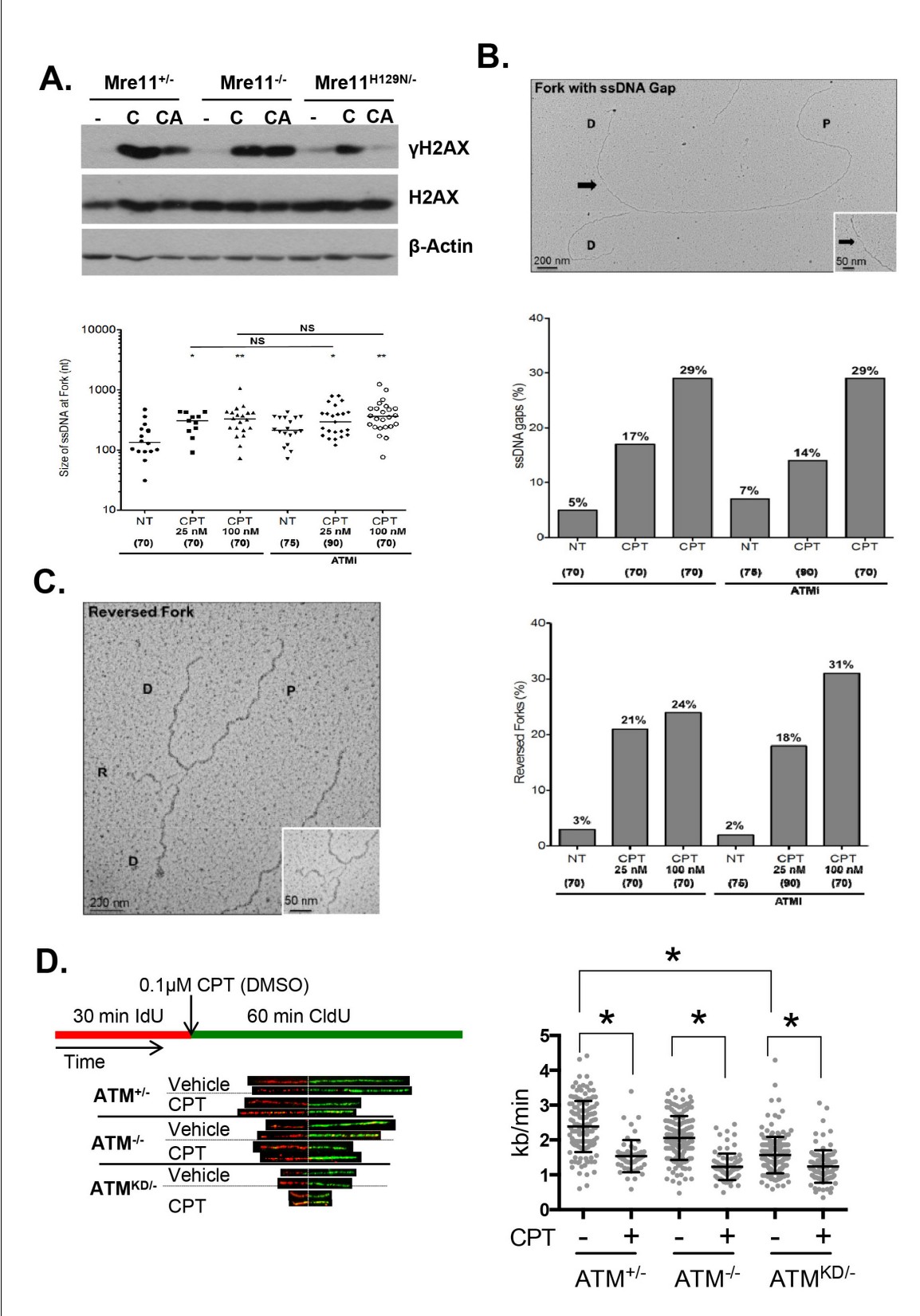

**Figure 7.** Atm-KD physically blocks the cleavage of CPT-stalled replication fork independent of fork reversal. (**A**) Western blots of *Mre11*[+/-], *Mre11*[-/-] or *Mre11*[H129N/-] (refereed as Nuc/- in the text) MEFs treated with CPT (C, 0.1 μM, 2 hr) or pretreated with ATM kinase inhibitor (CA, ATMi 15 μM, 1 hr,

*Figure 7 continued on next page*

*Figure 7 continued*

then together with CPT 0.1 µM, 2 hr). (B) Visualization and quantification of ssDNA at replication fork junctions in not treated (NT) cells and upon treatment with CPT with and without the ATM inhibitor (15 µM). Statistical analysis t-test according to Mann–Whitney results are *p≤0.1; **p≤0.01; ***p≤0.001. (C) Representative images and quantification of fork reversal by electron microscopy upon CPT (25 or 100 nM) and CPT+ATMi (Ku55933, 15 µM) treatment. P: parental DNA; D: daughter DNA strands, and R: reversed fork DNA. 70 replication forks were quantified for each genotype at each condition. Statistical analysis *t* test according to Mann–Whitney results are *p≤0.1; **p≤0.01; ***p≤0.001. (D) Representative images and quantification of the DNA fiber assay. Cells were incubated 30 min with 25 µM IdU, followed by 250 µM of CldU along with either vehicle (DMSO) or CPT (1 µM). The dot plot represents the summary of two independent experiments. *p<0.0001.

The following figure supplement is available for figure 7:

**Figure supplement 1.** Replication fork analyses in cells treated with CPT.

induced γ-H2AX described for $Atm^{KD/-}$ cells could be used as a biomarker to assess the function of ATM mutations, including the potential confounding MRN status.

While the lack of CPT-induced DSBs might explain the lack of CPT-induced Rad51 foci and SCEs in $Atm^{-/KD}$ cells, the reason for which I-SceI nuclease induced 'clean' DSBs could not be efficiently repaired by HR in the $Atm^{KD/-}$ cells is not apparent. The efficient accumulation of IR induced Rad51 foci in $Atm^{-/KD}$ cells, suggests that Atm-KD does not block the loading of Rad51 to resected DSBs, an early step required for HR (*Figure 6*). However, Rad51 loading or filament stability at a stalled replication fork might also be regulated differently from those at resected DSBs (from IR) and might occur independent of DSBs (*Schlacher, 2011*). In any case, our findings suggest that ATM-KD probably suppresses a later step of HR, especially holiday junction resolution, because the same structure specific nucleases (MUS81 and SLX1/4), are implicated in both holiday junction resolution and strand cleavage during Interstrand Crosslink (ICL) or Top1cc repair (*Zhang and Walter, 2014*; *Rouse, 2009*). The lack of MUS81 or SLX1/4 function could explain both the HR defects and the Top1cc removal defects in $Atm^{-/KD}$ cells. Accordingly, we found that $Slx4^{-/-}$ and, to a lesser extent $Mus81^{-/-}$, MEFs also display reduced numbers of CPT-induced DSBs (*Figure 7—figure supplement 1G*). In addition, the slow replication fork progression in $Atm^{-/KD}$ cells (*Figure 7D*) might also indirectly reduce HR efficiency measured by DR-GFP.

Intermolecular autophosphorylation is a common feature of ATM, DNA-PKcs and ATR (*Liu et al., 2011*; *Dobbs and Tainer, 2010*; *Jiang et al., 2015a*; *Bakkenist and Kastan, 2003*). We previously reported that DNA-PKcs recruited by Ku to DSBs block NHEJ in the absence of autophosphorylation (*Jiang et al., 2015a*). Here, ATM-KD recruited by MRE11 blocks replication fork cleavage upon collision with Top1cc in a remarkably similar manner, suggesting that autophosphorylation of PIKKs might have a common function in promoting their release from their respective activation partners (Ku for DNA-PKcs, MRN for ATM) and license distinct repair events (NHEJ for DNA-PKcs, strand cleavage for ATM). In the $Atm^{KD/-}$ cells, the defects in Top1cc removal combined with the lack of ATM kinase activity-mediated DNA damage response leads to severe genomic instability, embryonic lethality, and enhanced oncogenic potential.

## Materials and methods

### Genomic analyses of ATM mutations

ATM somatic mutations were collected from 5,402 cancer patients from 24 tumor types in TCGA, including ACC, BRCA, CHOL, COADREAD, ESCA, HNSC, KIRC, LAML, LIHC, MESO, PAAD, PRAD, SARC, STAD, THYM, UCS, BLCA, CESC, COAD, DLBC, GBM, KICH, KIRP, LGG, LUAD, OV, PCPG, READ, SKCM, TGCT, UCEC, and UVM. All maf files of those tumor types were downloaded using firehose (http://gdac.broadinstitute.org). To unify the annotation format, we first used the liftover tool (https://genome.ucsc.edu/cgi-bin/hgLiftOver) to map all coordinates to hg19, and then re-annotated variant effect with VEP (http://www.ensembl.org/info/docs/tools/vep/) under transcript NM_000051.3. Mutation types Missense_Mutation, In_Frame_Del, and In_Frame_Ins were classified as missense, while Nonsense_Mutation, Frame_Shift_Del, Frame_Shift_Ins, and Splice_Site were

classified as truncating mutations. Germline ATM mutations are from A-T patients in the Leiden open variable database LOVD2 (http://chromium.lovd.nl/LOVD2/home.php?select_db=ATM). Protein mutation position was curated and annotated based on the mutation effect provide by LOVD2. Start_Codon, Frame_Shift_Del, Large_DEL, Nonsense_Mutation, Frame_Shift_Ins, Splice_Site, and Stop_Codon are labeled as Truncating in our analysis.

## Mutation density estimation

To measure the mutation density of different ATM positions, we applied a Gaussian kernel smoother to smooth the number of mutations of each amino acid site. If we use $y(x_i)(i = 1, 2, \cdots, n)$ to represents the number of mutations in each site $x_i$, then the Gaussian kernel smoother of $y(x)$ is defined by:

$$y(x_i) = \frac{\sum_{j=1}^{n} K(x_i, x_j) y(x_j)}{\sum_{j=1}^{n} K(x_i, x_j)}$$

where $K(x_i, x_j) = \exp\left(-\frac{(x_i - x_j)^2}{2b^2}\right)$ is the Gaussian kernel, and $b$ indicates the window size (we use 80 in the current study). Missense mutations and truncating mutations were separately considered. To estimate the expected number of mutations in ATM, we assume the ratio between silent and non-silent mutation (e.g. missense mutation) is constant, and then the frequency, as well as its confidence interval, to observe a missense mutation can be estimated by fitting a binomial distribution with $n$ trials. The shadowed curve is the 95% confidence interval of fold change between observation and expectation after a Bonferroni Correction. The supplementary table 1 shows the result of the permutation analyses for the expected and observed mutations in ATM or ATM kinase domain (AA2711-2962, *Supplementary file 1B*).

## Structure simulation analyses

Homology model of human ATM was generated based on the crystal structure of human mTOR(PDB 4JSP) (*Yang et al., 2013*) using the ROBETTA server, setting default parameters (*Kim et al., 2004*; *Katyal et al., 2014*). Briefly, the AAs 2520–3056 of human ATM was submitted to the ROBETTA server and the mTOR crystal structure was identified as a template. After aligning the query sequence with the parent structure, a template was generated and the variable regions were modelled in the context of the fixed template using Rosetta fragment assembly. Finally, the model was subjected to several rounds of optimization by Rosetta's relax protocol. All structure analyses and visualizations were performed with pymol (http://pymol.org/).

## Human cancer cell lines

Human cancer cell lines – BT-474 (Breast cancer), Granta519 (MCL), HEC251 (Endometrium tumor) and SNU283 (Colon Cancer) were selected, since the CCLE (Broad Institute) have identified missense mutations in or around the ATM kinase domains in these cell lines. We have verified the identity of these cell lines to the best of our knowledge – based on ATM mutation status, ATM expression and activities (*Figure 1—figure supplement 1E*). HBL2 cell lines were selected as the MCL control, since similar to Granta519, HBL2 has deregulated expression of CyclinD1 and is WT for ATM. Other cell lines (Hela and U2OS) were also validated for normal expression of ATM and ATM kinase activity. None of the used cell lines were in the list of commonly misidentified cell lines maintained by the International Cell Line Authentication Committee (2016 May version).

## Mouse alleles and in vivo lymphocyte analyses

All the mouse alleles used have been previously described (*Yamamoto et al., 2012*; *de Boer et al., 2003*; *Zha et al., 2008*; *de Luca et al., 2005*), $Atm^C$, $Atm^{KD}$, $VavCre^+$, $Rosa^{ERCre}$. All the animal work was approved by and performed according to the regulations of the Institutional Animal Care and Use Committee (IACUC) of Columbia University. For development analyses, VN, VKD and littermate matched control mice were euthanized at 4–6 weeks of age. Total lymphocyte counts, flow cytometry analyses and in vitro class switch recombination were performed as described previously (*Li et al., 2008*). Briefly, splenic B and T-cells were isolated using Magnetic-Activated Cell Sorting (MACS) for CD43$^-$ and CD43$^+$ fractions, respectively, following the manufacturer's protocol (Miltenyi

Biotec). B-cells (CD43⁻) were stimulated with LPS (20 µg/ml) and IL-4 (20 ng/ml) for 4.5 days at a cellular concentration no more than $1 \times 10^6$ cells/ml, and the T-cells (CD43⁺) were activated by Con A (2 µg/ml) for 3.5 days (*Li et al., 2008*). Metaphases were collected at the end of the cytokine stimulation period after 4 hr of colcemid (100 ng/ml KaryoMax, Gibco) treatment, and stained with Telomere FISH probes as previously described (*Yamamoto et al., 2012*; *Franco et al., 2006*). Lymphoma- bearing animals were identified based on enlarged lymph nodes (B cell lymphomas) or severely hunched postures (thymic lymphomas) and analyzed. For the tumor studies, the investigators were not blinded.

## Mouse embryonic fibroblast (MEF)

MEFs were harvested at embryonic day 14.5 (E14.5) based on timed breeding and immortalized using retrovirus expression of the large and small SV40 T antigen. Immortalized culture was established after three passages (1:3 dilutions) after the primary infection. To induce 4-hydroxytamoxifen (4OHT) dependent nuclear translocation of ER-Cre recombinase (*Figure 3—figure supplement 1A*), the cells were plated at low density (~30% confluence), and passed three times (48 hr interval) in the presence of 4OHT (200 nM). Complete deletion of the conditional allele was confirmed using PCR as described previously (*Yamamoto et al., 2012*). For drug sensitivity assays, the immortalized MEF treated for 4OHT (3 rounds) were seeded in gelatinized 96-well plates ($6 \times 10^3$/well). Twenty-four hours after the initial seeding, the cells were treated with the compounds at indicated concentrations for 48 hr. The cell number was quantified using CyQuant (Molecular Probes), a nucleotide stain, per the manufacturer's instructions. The relative survival was calculated relative to the cell number in the untreated wells.

## Immunofluorescence and quantification of repair protein foci

Immunofluorescence was performed on 4% formaldehyde/PBS fixed, 1% Triton-X/PBS treated cells using the following antibodies: Rad51 (Clone PC130, 1:200, Calbiochem) phospho-RPA pT21 (Cat. # ab109394, 1:500, Cell Signaling) γH2AX (Cat. #07–164, 1:500, EMD Millipore), and CyclinA2 (Clone CY-A1, 1:500, Sigma-Aldrich). Slides were scanned on the Carl Zeiss Axio Imager Z2 equipped with a CoolCube1 camera (Carl Zeiss, Thornwood, NY). Metafer 4 software (MetaSystems, Newton, MA) was used for automated quantification of Rad51, pRPA, γH2AX foci. More than 500 cells were quantified for every experimental replicate. Images were exported and processed on ISIS software (MetaSystems).

## Western and southern blotting

The following antibodies were used for Western blots on whole cell lysate: ATM (clone MAT3, 1:5000), β-Actin (clone A5316, 1:20,000) and Vinculin (clone V284, 1:10,000) from Sigma-Aldrich; phospho-Kap1 (Cat. #A300-767A, 1:1000) and Top-I (Cat. # A302-590, 1:1000) from Bethyl laboratory; phospho-RPA pT21 (Cat. # ab109394, 1:5000) from Abcam; RPA (clone RPA30, 1:5000), γH2AX (Cat. #07–164, 1:1000) and H2AX (Cat. #07–627, 1:1000) from EMD Millipore; phospho-Chk1 (clone 133D3, 1:500), Mre11 (Cat. #4895S, 1:1000), Kap1/TIF1β (clone C42G12, 1:1000), and Pten (clone D4.3 XP, 1:1000) from Cell Signaling and PCNA (clone PC10, 1:1000) from Santa Cruz. To confirm deletion of the ATM conditional allele in VKD and VN mouse models, Southern blot was performed on KpnI digested genomic DNA and probed with the ATMCKO 3' probe (*Zha et al., 2008*). For clonal analyses of the thymic lymphomas or the B cell lymphomas, Southern blot was performed on EcoRI-digested genomic DNA, and blotted with Jβ1.6, Jβ2.7 probes (*Zha et al., 2008*; *Khor and Sleckman, 2005*) or Jh4 and myc-A probe (*Gostissa et al., 2009*).

## Sister chromatid exchange assay

Cells were incubated for two doubling times (48 hrs) with Bromodeoxyuridine (BrdU, Sigma-Aldrich, 5 µg/ml). During the second doubling period (hours 25–48), the cells were also incubated with either vehicle (DMSO) or CPT (1 µM). BrdU-incorporated sister chromatids were quenched using Hoechst33258 (50 µg/ml) treatment followed by UV exposure, then stained with DAPI (Vectashield). Slides were scanned for metaphase spreads using the Carl Zeiss Axio Imager Z2 equipped with a CoolCube1 camera (Carl Zeiss, Thornwood, NY) and Metafer 4 software (MetaSystems, Newton,

MA). Spreads were quantified and images exported via ISIS software (MetaSystems). Over 40 metaphases were quantified for each genotype. A total of three independent replicates were performed.

## COMET assay
Neutral and alkaline COMET assays were performed per manufacturer's protocol (Trevigen, Gaithersburg, MD). Slides were scanned for COMETs using Metafer 4 and tail lengths quantified using ISIS.

## In-vitro complex of enzymes (ICE) assay
Covalent complexes of TopI-DNA were detected as previously described (*Nitiss et al., 2012*). Briefly, cells were gently lysed in 1% Sarkosyl/TE solution, ultracentrifuged through a $CsCl_2$ gradient (6 M) and the resulted genomic DNA was dissolved in TE. Equivalent amounts of DNA were loaded on 0.45 µm Nitrocellulose membranes, and probed for Top-1 (Cat. # A302-590, 1:1000, Bethyl). Loading controls were probed with radiolabeled total mouse genomic DNA.

## DNA fiber assay
Cells were first incubated with 25 µM IdU (30 min) then with additional 250 µM CldU (Sigma), trypsinized and harvested in ice-cold 1 x PBS, lysed using 1% SDS/Tris-EDTA and stretched along glass coverslips. Slides were stained using primary Anti-IdU (B44, BD Biosciences), CldU (BU1/75 (ICR1), Abcam) antibodies and corresponding secondary anti-Rat Alexa594, and anti-mouse Alexa488 (Molecular Probes) antibodies. DNA fibers were analyzed on the Nikon Eclipse 80i microscope equipped with remote focus accessory and CoolSNAP HQ camera unit using the 60 x /1.30 NA oil Plan Fluor lens. All images were processed and quantified with NIS-Elements AR. Lengths of fibers were calculated with a base pair length of 3.4 Å (340 pm).

## Comparative genomic hybridization (CGH) assay
1 µg of differentially labeled genomic DNA from tumor and non-tumor control (kidney) derived from the same animal were hybridized on the 244 k Mouse Genome CGH microarray platform (Agilent) and analyzed on Agilent Genomic Workbench and Microsoft Excel as previously described (*Zha et al., 2010*; *Yamamoto et al., 2015*).

## Electronmicroscopic (EM) analyses of genomic DNA
In vivo psoralen cross-linking, isolation of genomic DNA from mammalian cells, enrichment of replication intermediates, and analysis of data was performed as previously described (*Neelsen et al., 2014*; *Thangavel et al., 2015*). Briefly, 5–10 × 10^6 Mouse Embryonic Fibroblasts (Rosa^+/ERCRE Atm^+/C) cells were harvested and genomic DNA was cross-linked by three rounds of incubation in 10 µg/ml 4,5',8-trimethylpsoralen (Sigma-Aldrich) and 3 min of irradiation with 366 nm UV light on a precooled metal block. Cells were lysed and cellular proteins were digested with 2 mg proteinase K (Life technologies) in 5 ml digestion buffer. DNA was purified by isopropanol precipitation, restriction digested with PvuII HF for 4 hr at 37°C, and replication intermediates were enriched using benzolylated naphthoylated DEAE-cellulose (Sigma–Aldrich) in 3 ml BioRad poly-prep chromatography columns. Samples were prepared by spreading DNA on carbon-coated grids in the presence of benzyl-dimethyl-alkylammonium chloride and formamide. Rotary platinum shadowing was performed using the Balzers BAF400 with Quartz crystal thin film monitor. Images were acquired on a JOEL 1200 EX transmission electron microscope with side-mounted camera (AMTXR41 supported by AMT software v601) and analyzed with ImageJ (National Institutes of Health).

## In vivo isogenic Leukemia and irinotecan treatment (diagrammed in *Figure 4A*)
To determine the therapeutic effect of clinically used Topo1 inhibitor – irinotecan in vivo, we generated isogenic leukemia by transducing bone marrow from Rosa^+/CreERT2 Atm^C/KD mice treated with 5-Fluorouracil (5-FU) were transduced with retrovirus encoding activated Notch1 (ΔE-Notch1) and GFP(MSCV-ΔENOTCH-IRES-GFP) (*Tzoneva et al., 2013*) and transplanted into semi-lethal irradiated (8.5 Gy) recipient mice (n = 3). Once the leukemia developed (monitored by peripheral blood GFP+ cells), ~1 × 10^6 GFP+ leukemia cells (from one mouse to ensure clonality) were transduced to 2nd

recipients (n = 4–6). 4 days later, the 2nd recipients were fed with oral tamoxifen (n = 3) or carrier alone (sun flower oil, n = 3) for two days. When the 2nd recipients developed leukemia, the leukemia cells were genotyped for ATM status (KD/- or KD/C) and analyzed for clonality (TCRβ rearrangements by Southern). Once the isogenic (defined by identical TCRβ rearrangements) leukemia was obtained, they were transduced to tertiary recipients (n = 5–10 per genotype). 3 days after the transplantation, the tertiary recipients were treated with IP injection of irinotecan (10 mg/kg) for 5 days or vehicle alone (for 5 days) and observed for leukemia development. In ~10 days after the last drug injection, the vehicle groups displayed terminal diseases and both the vehicle control and the irinotecan treated mice were euthanized and analyzed for GFP+ cells in the spleen, histology and spleen weight and size. Similar experiments were performed on $Rosa^{+/CreERT2}$ $Atm^{C/C}$ bone marrow derived, isogenic $Atm^{C/C}$ vs $Atm^{-/-}$ Notch driven leukemia and representative results are shown in *Figure 4—figure supplement 1C–F*.

## Acknowledgements

We thank Dr. Richard Baer for their comments and critical review of the manuscript. We thank Drs. Agata Smogorzewska, Alex Bishop, Laura Niedernhofer, David O Ferguson and Richard Baer for providing immortalized murine embryonic fibroblast cells critical for this study and Drs. John Tainer and Jean Gautier for providing Mre11 nuclease inhibitors and Drs. Peter J McKinnon and Tony Huang for sharing experimental protocols. We apologize to colleagues whose work could not be cited due to space limitations and was covered by reviews instead. This work is in part supported by NIH/NCI 5R01CA158073, 5R01CA184187, 1P01CA174653-01 and American Cancer Society Research Scholar Grant (RSG-13-038-01 DMC) to SZ, 1R01CA185486-01, 1R01CA179044-01A1 and U54 CA193313 to RR, NIH/NIGMS GM102362 to DW and NIH R01GM108648 to AV. SZ is the recipient of the Leukemia Lymphoma Society Scholar Award. JW is also supported by Precision Medicine Fellowship (UL1 TR000040). LS and WJ are supported by NIH/NCI T32-CA09503.

## Additional information

### Funding

| Funder | Grant reference number | Author |
|---|---|---|
| National Cancer Institute | UL1 TR000040 | Jiguang Wang |
| National Cancer Institute | T32-CA09503 | Lisa Sprinzen<br>Wenxia Jiang |
| National Institute of General Medical Sciences | NIH R01GM108648 | Alessandro Vindigni |
| National Institute of General Medical Sciences | NIH/NIGMS GM102362 | Dong Wang |
| National Cancer Institute | U54 CA193313 | Raul Rabadan |
| National Cancer Institute | 1R01CA185486-01 | Raul Rabadan |
| National Cancer Institute | 1R01CA179044-01A1 | Raul Rabadan |
| American Cancer Society | RSG-13-038-01 DMC | Shan Zha |
| National Cancer Institute | 1R01CA158073 | Shan Zha |
| Leukemia and Lymphoma Society | Scholar Award | Shan Zha |
| National Cancer Institute | 5R01CA184187 | Shan Zha |
| National Cancer Institute | 1P01CA174653 | Shan Zha |

The funders had no role in study design, data collection and interpretation, or the decision to submit the work for publication.

## Author contributions
KY, SZ, Designed experiments, Wrote the paper, Generated and analyzed the Atm KD mice and cells, Acquisition of data, Analysis and interpretation of data, Contributed unpublished essential data or reagents; JW, Performed the bioinformatics analyses of human ATM mutations and wrote the related sections, Acquisition of data, Contributed unpublished essential data or reagents; LS, Designed experiments, Generated and analyzed the Atm KD mice and cells, Acquisition of data, Analysis and interpretation of data, Drafting or revising the article, Contributed unpublished essential data or reagents; JX, Performed structural modeling for ATM kinase and FATC domain and provided the related figures, Acquisition of data, Analysis and interpretation of data, Drafting or revising the article, Contributed unpublished essential data or reagents; CJH, Performed the electron microscopy analyses of the replication forks and generated the related figures, Acquisition of data, Contributed unpublished essential data or reagents; CL, DGL, WJ, Generated and analyzed the Atm KD mice and cells, Acquisition of data, Analysis and interpretation of data, Contributed unpublished essential data or reagents; BJL, Generated and analyzed the Atm KD mice and cells, Acquisition of data, Analysis and interpretation of data, Drafting or revising the article, Contributed unpublished essential data or reagents; AV, Designed experiments, Performed the electron microscopy analyses of the replication forks and generated the related figures, Drafting or revising the article, Contributed unpublished essential data or reagents; DW, Designed experiments, Performed structural modeling for ATM kinase and FATC domain and provided the related figures, Analysis and interpretation of data, Drafting or revising the article, Contributed unpublished essential data or reagents; RR, Designed experiments, Performed the bioinformatics analyses of human ATM mutations and wrote the related sections, Contributed unpublished essential data or reagents

## Author ORCIDs
Shan Zha, http://orcid.org/0000-0002-6568-1818

## Ethics
Animal experimentation: All the animal work was approved by and performed according to the regulations of the Institutional Animal Care and Use Committee (IACUC) of Columbia University (protocol no: AAAF7653, AAAD6250, AAAJ3651).

## Additional files

### Supplementary files
• Supplementary file 1. Nonsynonymous mutations reported for A-T patients (A) in the Leiden open variable database (http://chromium.lovd.nl/LOVD2/home.php?select_db=ATM) or The Cancer Genome Atlas (TCGA) at the time of primary analyses (5402 cases, (B). At the time of primary analyses, 2 TCGA cancer cases (1 and 2 in the table below) have mutations involving N2875 of human ATM. Among them, the first case also has 'shallow del' in ATM region, consistent with Mut/- genotype. Since then, one additional prostate cancer (case 3) was reported to have N2875S mutations. While ATM region deemed 'diploid', the allele frequency of the N2875S allele is 0.92 (92% of the reads in this region are mutated), consistent with phenotypically - homozygous status. 1) TCGA-EW-A1J6-01 breast cancer N2875S shallow del; 2) TCGA-G9-7521-01 prostate cancer N2875K diploid; 3) TCGA-YL-A8S9-01 prostate cancer N2875S diploid; allele freq = 0.92.

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
