## [Decision Letter]

Thank you for submitting your article "Kinase-dead ATM protein is highly oncogenic and can be preferentially targeted by Topo-isomerase I inhibitors" for consideration by *eLife*. Your article has been favorably evaluated by Charles Sawyers (Senior editor) and three reviewers, one of whom is a member of our Board of Reviewing Editors.

The reviewers have discussed the reviews with one another and the Reviewing Editor has drafted this decision to help you prepare a revised submission.

In previous studies these authors made the rather surprising finding that while mice deficient in ATM are viable, mice that express a kinase dead version of ATM exhibit early embryonic lethality. This suggests that the loss of ATM and expression of a kinase dead version of ATM leads to very different cellular defects. In this manuscript they explore the potential function of ATM mutations in cancer. They find that there is an increase in cancers with mutations that would be expected to lead to the generation of a kinase-dead version of ATM, suggesting that this mutation may promote cancer. They show that the expression of a kinase dead version of ATM, but not loss of ATM, prevents the removal of Topo-isomerase I-DNA adducts making these tumor cells more sensitive to Topo-isomerase inhibitors. Overall the experiments are well done and the findings shed important light on the potential unique activity of the kinase dead version of ATM. Moreover, they are relevant to the treatment of cancers with ATM mutations that effect kinase activity.

The following issues must be addressed before publication:

Main Points:

1) Given that the effects on CPT-induced DSB formation are not observed in ATM null cells, it appears that the effects in KD mutants represent a dominant negative effect and not a normal function of ATM in the response to DSBs. Therefore, the claim that they have identified a new function of ATM is not valid.

2) The identification of N2875K mutation, which is embryonic lethal in a mouse in the absence of wild-type ATM, in two human tumors in the TCGA is potentially important. But the authors need to specify if the N2875K in the human tumors was homozygous or additionally associated with loss of chromosome 11q23. The selective sensitivity of ATM kinase dead human cancer cell lines to camptothecin could be demonstrated by knockout of the kinase dead protein. This would considerably strengthen this experiment with the human cancer cell lines.

3) In Figure 4—figure supplement 1 and Figure 5 pRPA32(T21) was used to show in all genotypes, ssDNA formation was not significantly affected upon CPT treatment (Western blot); however, number of Rad51 foci was significantly reduced in *ATM^KD/-^*cells. The authors attributed this phenotype to lack of DSBs in *ATM^KD/-^*cells. Another explanation would be defect in Rad51 loading in *ATM^KD/-^*cells assuming equivalent amount of ssDNA is present in all cells. While reduce number of DSBs might explain why Rad51 foci decreased in *ATM^KD/-^*cells, how does it explain equivalent levels of ssDNA in all three genotypes? The authors should include a more detailed discussion.

4) In Figure 5, it was shown that ATM-KD protein blocks CPT-induced DSB formation in MRN-dependent manner. The authors concluded that MRN (with nuclease active or inactive MRE11) recruited *ATMKD* to DNA and ATM-KD protein blocks the conversion of Top1cc lesion to DSBs. However, for this to occur, MRN has to recognize Top1-induced SSBs first in order to recruit ATM. Have there been reports that MRN recognizes and binds to SSBs?

5) In the same figure, it looks like upon CPT treatment, ATM inhibitor in cells expressing nuclease-deficient MRE11 abolished gH2AX formation (indicative of DSB generation) while in wild type MRE11-expressing cells only partially reduced gH2AX. Is the difference reproducible? If so, this would suggest that MRE11 nuclease also contribute gH2AX (DSB) formation in CPT-treated *ATM^KD/-^*cells (again, if MRN binds to SSB). What is the likely explanation?

Other Specific Issues (in no particular order):

1) The graph in Figure 2 indicates that the incidence of Thymic lymphoma is about 65%, not 75% as indicated in the text. The difference between the VN and VKD animals is now not very different. Also, were tests for statistical significance performed on all these comparisons?

2) To be convincing, rather than showing four examples, the spleen weight of appropriate numbers of mice should be reported for Figure 3—figure supplement 1.

3) The results in Figure 4 (particularly the comparison with *Atm^-/-^*) need to be quantified in multiple experiments to be convincing.

4) Why is there no effective suppression of CPT-induced DSB formation in the *Mre11^+/-^* cells in Figure 5?

5) The ATM kinase dead mutation generated by Canman et al. contained two missense mutations, not a single N2875K mutation.

6) The nomenclature for the mice needs to be clarified in the Abstract and Introduction. The meaning of *Atm^C^* is not clear in the Abstract.

7) There are grammatical errors that should be corrected.

8) Work by Katyal S et al. (published in 2014 on Nature Neuroscience), which was referenced by the authors, showed that ATM depletion inflicted more DNA damage upon CPT treatment than inhibition of ATM kinase activity in quiescent astrocyte. However, the authors reported here that *Atm^KD/-^*cells are more sensitive to CPT than *Atm^-/-^* cells. What may contribute to this difference? Cell type, cell cycle (proliferating vs. quiescent) or others.

[Editors' note: further revisions were requested prior to acceptance, as described below.]

Thank you for resubmitting your work entitled "Kinase-dead ATM protein is highly oncogenic and can be preferentially targeted by Topo-isomerase I inhibitors" for further consideration at *eLife*. Your revised article has been favorably evaluated by Charles Sawyers (Senior editor), a Reviewing editor, and two reviewers.

The manuscript has been improved but there are some remaining issues that need to be addressed before acceptance, as outlined below:

First, in the current Figure 7, the authors showed that Mre11 nuclease activity (*Mre11^+/-^)* contributes to CPT-induced DSB generation in the presence of ATM-KD protein. Two additional biological replicates were provided in Figure 7—figure supplement 1. However, the data presented in the supplemental figures are not convincing – there is no more gH2AX in *Mre11^+/-^* than in *Mre11*^H129N/-^. There is a very little gH2AX induction in these two additional experiments, compared to Figure 7. Therefore, it is not straightforward to conclude whether the nuclease activity of Mre11 also contribute to CPT-induced DSB generation in this context. The author should re-phase accordingly.

The second point regards the lack of RAD51 foci in *ATM^KD^* cells treated with CPT. The authors attributed this observation to the lack of DSB generation *ATM^KD^* cells, rather than impaired RAD51 loading. While this statement is valid with the support the absence (or reduced number) of gH2AX foci and neutral comet assay, the possibility of defective RAD51 loading should not be excluded. The authors stated in the response to reviewer's comment that the ssDNA at stalled replication fork would not lead to RAD51 loading. However, two independent studies have directly or indirectly shown that RAD51 is present at stalled replication forks to promote fork reversal or protect the integrity of nascent DNA strands (Zellweger R et al., JCB, 2015 and Schlacher K et al., Cell, 2011). The later study actually proposed that the stability of RAD51 filament, rather than RAD51 "loading", is important for protecting nascent DNA at stalled fork. Whether RAD51 loading is an issue or not does not affect the core value of this manuscript, but it would be nice if the authors could make a minor revision to include all possibilities.

---

## [Author Response]

*The following issues must be addressed before publication:*

*Main Points:*

1) Given that the effects on CPT-induced DSB formation are not observed in ATM null cells, it appears that the effects in KD mutants represent a dominant negative effect and not a normal function of ATM in the response to DSBs. Therefore, the claim that they have identified a new function of ATM is not valid.

We would like to thank the reviewer for pointing this out. We now stated in the Discussion (first paragraph) as we identified a potential “neomorphic” oncogenic function of kinase-dead ATM protein. We also noted that, given our results from both *Atm_KD/-_*cells and ATM kinase inhibitor treated cells, suggests that both ATM-KD and the “normal” ATM protein are recruited to CPT induced stalked replication forks by an MRE11-dependent mechanism (see response to point 4 for Mre11 binding specificities). And in this context, ATM kinase activity and potentially the inter-molecular auto-phosphorylation, are necessary to release ATM, which in turn allows fork cleavage. As such, our ATM-KD model likely reveals a phosphorylation dependent structural function of normal ATM protein that is not apparent in the ATM null mouse/cells (lacking both ATM protein and its kinase activity). We have now clarified this point in the first paragraph of the Discussion.

2) The identification of N2875K mutation, which is embryonic lethal in a mouse in the absence of wild-type ATM, in two human tumors in the TCGA is potentially important. But the authors need to specify if the N2875K in the human tumors was homozygous or additionally associated with loss of chromosome 11q23. The selective sensitivity of ATM kinase dead human cancer cell lines to camptothecin could be demonstrated by knockout of the kinase dead protein. This would considerably strengthen this experiment with the human cancer cell lines.

At the time of primary submission, 2 TCGA cancer cases (1 and 2 in the table below) have mutations involving N2875. Among them, the first case also has “shallow del” in ATM region, consistent with Mut/- genotype. Since then, one additional prostate cancer (case 3) was reported to have N2875S mutations. While ATM region deemed “diploid”, the allele frequency of the N2875S allele is 0.92 (92% of the reads in this region are mutated), consistent with phenotypically – homozygous status (Results and [Supplementary-material SD1-data]).

1) TCGA-EW-A1J6-01breast cancerN2875Sshallow del;2) TCGA-G9-7521-01prostate cancerN2875Kdiploid;3) TCGA-YL-A8S9-01prostate cancerN2875Sdiploidallele freq= 0.92;

To address whether the hypersensitivity to CPT in human cancer cell lines expressing catalytically inactive ATM depending on the expression of ATM protein, we used shRNA to knockdown ATM in ATM mutated Granta519 cells, a human mantle cell lymphoma (MCL) cell line, and a control ATM WT MCL cell line – HBL2. While knockdown of ATM in HBL2 did not significant alter the CPT sensitivity, knockdown of mutated ATM in Granta519 cells moderately yet significantly reduced CPT sensitivity (new Figure 3). Moreover, we showed that the CPT hyper-sensitivity in Granta519 cells is associated with increased apoptosis (measured by PI staining) and shRNA knockdown of the mutated ATM led to statistic significant decrease of CPT (5nM 48hr) induced apoptosis in Granta519 cells without significant changes in the baseline (without CPT) apoptosis cell percentage (New Figure 3—figure supplement 1).

3) In Figure 4—figure supplement 1 and Figure 5 pRPA32(T21) was used to show in all genotypes, ssDNA formation was not significantly affected upon CPT treatment (Western blot); however, number of Rad51 foci was significantly reduced in ATM^KD/-^cells. The authors attributed this phenotype to lack of DSBs in ATM^KD/-^ cells. Another explanation would be defect in Rad51 loading in ATM^KD/-^cells assuming equivalent amount of ssDNA is present in all cells. While reduce number of DSBs might explain why Rad51 foci decreased in ATM^KD/-^ cells, how does it explain equivalent levels of ssDNA in all three genotypes? The authors should include a more detailed discussion.

We noted that “IR” induced Rad51 foci formation (Now Figure 6 (previously Figure 4)) are not statistically reduced in *Atm_KD/-_*cells, in contrast to the severe reduction IR induced Rad 51 foci in BRCA1 mutant cells with known Rad51 loading defects. Together with the reduced CPT induced Rad51 foci, these data indicate that ATM-KD protein does *not* block RAD51 loading from clean DSBs or RPA coated single strand DNAs. We have now clarified this in the Results (subsection “ATM-KD blocks replication-dependent removal of Topo-isomerase I DNA adducts at the step of strand cleavage” second paragraph).

In the event of “stalled replication”, RPA coated ssDNA (marked by phosphorylated RPA) can accumulate due to the un-coupling of the leading and lagging strands, as diagrammed in Figure 5 (blue dashed box), observed in electron-microscopy (EM) experiments in Figure 7. This accumulation of single stand DNA at the stalled replication forks were thought to be the precursor for fork reversion and could be generated before DSBs and would NOT lead to Rad51 loading. This is distinct to from the pRPA bounded ssDNA originated from resected DSBs. We note, upon CPT treatment, *Atm_KD/-_*cells (or Atm kinase inhibitor treated cells) accumulates normal levels of pRPA (measured by Western Blotting, Figure 6, Figure 6—figure supplement 1), single strand DNA at the fork (by EM, Figure 7), fork reversal products (by EM, Figure 7) and delay in replication fork progression (by DNA fiber assay, Figure 7), together those data support that Atm-KD does *not* block fork-uncoupling or fork reversal.

Given the robust levels of IR induced RAD51 foci in *Atm_KD/-_* cells and the lack of CPT induced RAD51 foci, our results are most consistent with the lack of CPT induced DSBs due to strand cleavage defects.

This is further supported by the lack of CPT induced g-H2AX and neutral comet tails. We have now moved the diagram from supplementary Figure to Figure 5 to better elucidate our working model and complex structures at stalled replication forks. We also expanded the Results and discussions to better explain our rationales and our working model (subsection “ATM-KD blocks replication-dependent removal of Topo-isomerase I DNA adducts at the step of strand cleavage”).

4) In Figure 5, it was shown that ATM-KD protein blocks CPT-induced DSB formation in MRN-dependent manner. The authors concluded that MRN (with nuclease active or inactive MRE11) recruited ATMKD to DNA and ATM-KD protein blocks the conversion of Top1cc lesion to DSBs. However, for this to occur, MRN has to recognize Top1-induced SSBs first in order to recruit ATM. Have there been reports that MRN recognizes and binds to SSBs?

We thank the reviewers to point out these important references that were missing in the previous draft. Indeed, independent biochemistry experiments from several laboratories showed that purified MRE11 proteins from various species (human, yeast, etc.) binds to ssDNA as well as dsDNA (Nucleic Acids Res. 2001 Mar 15; 29(6): 1317–1325; Cell, 95 (1998), pp. 705–716); Genes Dev., 13 (1999), pp. 1276–1288). We have now included these references in our manuscript (subsection “ATM-KD blocks replication-dependent removal of Topo-isomerase I DNA adducts at the step of strand cleavage”, third paragraph).

Moreover, these experiments also revealed that the affinity of Mre11 to ssDNA is even higher than its affinity to dsDNA. Moreover, Mre11 binds to both linear and circular form of ssDNA, suggesting the binding between MRE11 to ssDNA does not require free DNA ends. This is likely the structure that is generated at stalled replication fork after the leading and lagging strand are uncoupled (see Figure 5 blue dash box).

Correspondingly, recent studies using the iPOND technology have identified MRE11 at the stalled forks in vivo(Sirbu et al., 2011). We have now discussed these important supporting references in the third paragraph of the subsection “ATM-KD blocks replication-dependent removal of Topo-isomerase I DNA adducts at the step of strand cleavage” and the Discussion section.

5) In the same figure, it looks like upon CPT treatment, ATM inhibitor in cells expressing nuclease-deficient MRE11 abolished gH2AX formation (indicative of DSB generation) while in wild type MRE11-expressing cells only partially reduced gH2AX. Is the difference reproducible? If so, this would suggest that MRE11 nuclease also contribute gH2AX (DSB) formation in CPT-treated ATM^KD/-^cells (again, if MRN binds to SSB). What is the likely explanation?

In current Figure 7 (previous Figure 5), ATM kinase inhibitors abolished CPT induced DSBs in Mre11 nuclease deficient cells more effectively than in *Mre11+/-* cells. We now provided two independent biological replicates of this experiments (current Figure 7—figure supplement 1). This result is robust and reproducible. Several possible explanations exist. First, MRE11 nuclease activity might help to cleave the DNA and therefore release the stalled Atm-KD (possibly together with MRN complex) from the replication forks and in turn “allow” CPT induced strand cleavage. In this context, lack of Sae2, a potent modulator of MRE11 nuclease activity, cause hyper activation of Tel1 (yeast orthology of ATM). Two recent papers have identified Mre11 mutants that suppresses persistent ATM (Tel1 in yeast) mediated checkpoint activation in Sae2 deficient cells (Proc Natl Acad Sci U S A. 2015 Apr 14;112(15):E1880-7; and EMBO J. 2015 Jun 3;34(11):1509-22.) (Discussion section), suggesting Mre11 and its nuclease activity might contribute to the termination of ATM signaling. Another not mutually exclusive possibility would be, Mre11 nuclease activity directly contribute to fork cleavage, which in turn resolves the stalled replication fork. In this context, we found that two Mre11 nuclease inhibitors – PFM01 and Mirin both reduced CPT-induced gH2AX in *Atm_+/+_* and *Atm_-/-_* cells. This data is now included in Figure 7—figure supplement 1 and discussed in the Discussion section. However, given the toxicity of the Mre11 nuclease inhibitors in general, further experiments would be necessary to solidify the role of MRE11 nuclease activity in CPT induced double stand breaks.

*Other Specific Issues (in no particular order):*

1) The graph in Figure 2 indicates that the incidence of Thymic lymphoma is about 65%, not 75% as indicated in the text. The difference between the VN and VKD animals is now not very different. Also, were tests for statistical significance performed on all these comparisons?

We apologize for this error. We mean that “the incidence of lymphomas (not thymic lymphomas) is 75% in VKD vs. 56% in VN”. We have now correct this. Since there is no B cell lymphomas in VN mice, we compared the kinetic of “thymic lymphomas” in VKD vs. in VN mice using Mantel-Cox/log rank test and the p=0.03 (subsection “Expression of Atm-KD is more oncogenic than loss of Atm”, second paragraph and Figure 2 legend).

2) To be convincing, rather than showing four examples, the spleen weight of appropriate numbers of mice should be reported for Figure 3—figure supplement 1.

We performed additional biological repeats for the isogenic *ATM_C/C_* and *ATM_-/-_* activated Notch- drive leukemia. The results supported our initial observation. The statistical analyses as well as the presentative histology figures are now included in Figure 4—figure supplement 1. Given the additional figure panels, we have moved the data related to the isogenic leukemia to Figure 4 (from Figure 3 in the original submission) and the subsequent figures were also shifted correspondingly.

3) The results in Figure 4 (particularly the comparison with ATM^-/-^) need to be quantified in multiple experiments to be convincing.

We have now performed two additional biological repeats on independent derived *Atm_KD/-_*and *Atm_-/-_* MEF lines. The results are shown in Figure 5—figure supplement 1. Two different exposures for each experiments were also shown. Given the intrinsic error of densitometry analyses (e.g.which exposure to use and how to subtract backgrounds), we felt that the differences are robust enough and three independent biological replicates would make this point stronger.

4) Why is there no effective suppression of CPT-induced DSB formation in the Mre11^+/-^ cells in Figure 5?

While Mre11_-/-_ is cell lethal and embryonic lethal, Mre11_+/-_ mice and cells have not measurable repair defects, suggesting one copy of Mre11 is sufficient to support normal function of Mre11 (Buis et al., 2008). We have now showed three biological replicates on this experiments (Figure 7 and Figure 7—figure supplement 1). Together, these experiments indicate that ATM kinase inhibitor does reduce CPT induced g-H2AX in Mre11_+/-_ cells.

5) The ATM kinase dead mutation generated by Canman et al. contained two missense mutations, not a single N2875K mutation.

We have now made this clear in the text (subsection “Cancer-associated ATM mutations are enriched for kinase domain missense mutations”). N2875K is one of the two mutations (N2875K and D2870A) engineered into the ATM KD allele in the Canman et al. paper.

6) The nomenclature for the mice needs to be clarified in the Abstract and Introduction. The meaning of Atm^C^ is not clear in the Abstract.

Given the 150 words limited for the abstract, we have now replaced *Atm_C/KD_*and *Atm_C/C_* in the Abstract with “Atm-proficient” and further define the nomenclature in the main text.

7) There are grammatical errors that should be corrected.

We have proofread the manuscript to minimize this issue.

8) Work by Katyal S et al. (published in 2014 on Nature Neuroscience), which was referenced by the authors, showed that ATM depletion inflicted more DNA damage upon CPT treatment than inhibition of ATM kinase activity in quiescent astrocyte. However, the authors reported here that ATM^KD/-^ cells are more sensitive to CPT than ATM^-/-^ cells. What may contribute to this difference? Cell type, cell cycle (proliferating vs. quiescent) or others.

Two studies, including the Katyal S et al. and an earlier study by Alagoz, M et Al. (Alagoz et al., 2013; Katyal et al., 2014), described defects in Top1cc metabolism in *Atm_-/-_* neuronal cells, especially astrocytes. We would like to point out that the Top1cc assay in current Figure 5 (previously Figure 4) and the biological repeats (current Figure 5—figure supplement 1) all showed that *Atm_-/-_* MEFs accumulates higher levels of Top1cc than *Atm_+/+_* cells, especially at high CPT dose (15µM), consistent with the results from astrocytes. In this context, cell type alone is insufficient to explain these differences.

What is unique to our study is, we found that *Atm_KD/-_*cells accumulate much higher levels of Top1cc than *Atm_-/-_* (and *Atm_+/+_*) cells. In addition to the high CPT dose used in previous studies (15µM, IC50 for CPT in WT cells is ~ 0.01uM), we showed that *Atm_KD/-_*cells accumulate Top1cc at 0.1µM CPT (Figure 5). At the mechanistic level, we identified that *Atm_KD/-_* cells, but not *Atm_-/-_* cells, have a defects in CPT induced DSB formation (gH2AX). Figure 4—figure supplement 1 shows that CPT induced DSBs formation occurs predominantly in S phase cells, while previous studies were primarily conducted in “quiescent astrocytes”, which would not uncover replication associated Top1cc removal defects. Based on this, we suggested that the difference is likely to be cell cycle dependent.

In this context, Katyal, S et al. suggests that *Atm_-/-_* cells have a defect in CPT induced Top1- proteolytic degradation astrocytes. Yet, we found that CPT induced degradation of full length Topo1 is not measurably affected in proliferating *Atm_KD/-_*or *Atm_-/-_* MEFs. Studies from *Xenopus* extracts showed that collision with replication forks induce proteolytic degradation of proteins covalently linked to DNA (e.g. Top1cc) (Duxin et al., 2014). While how ATM affects in CPT induced Top1 degradation in astrocytes are yet to be determined, the robust replication dependently proteolytic program might explain the lack of the proteolytic defects in proliferating *Atm_KD/-_*and *Atm_-/-_* MEFs in our study. Together, our data support a model in which ATM is recruited to CPT induced stalled replication forks, where auto-phosphorylation of ATM is necessary to license strand cleavage. In this case, the absence of ATM (*Atm^-/-^*) would not pose an inhibitory effect. Finally, since the cancer cells were preferentially targeted by “proliferation/replication” specific damages, the replication associated Top1cc removal defects in *Atm_KD/-_*cells uncovered in our study are particularly relative to cancer therapy. We now discussed this in detail in the subsection “ATM-KD blocks replication-dependent removal of Topo-isomerase I DNA adducts at the step of strand cleavage”.

[Editors' note: further revisions were requested prior to acceptance, as described below.]

*The manuscript has been improved but there are some remaining issues that need to be addressed before acceptance, as outlined below:*

*First, in the current Figure 7, the authors showed that Mre11 nuclease activity (Mre11+/-) contributes to CPT-induced DSB generation in the presence of ATM-KD protein. Two additional biological replicates were provided in Figure 7—figure supplement 1. However, the data presented in the supplemental figures are not convincing – there is no more gH2AX in Mre11+/- than in Mre11H129N/-. There is a very little gH2AX induction in these two additional experiments, compared to Figure 7. Therefore, it is not straightforward to conclude whether the nuclease activity of Mre11 also contribute to CPT-induced DSB generation in this context. The author should re-phase accordingly.*

We now emphasized – “further investigation is needed to clarify this issue”. But we also would like to point out that in the first revision of the manuscript – the possible role of the MRE11 nuclease activity is *only* mentioned in the Discussion and not presented as a “conclusion”. Specifically, in the Discussion, we cautiously stated that “…suggesting that Mre11 nuclease activity might contribute to the removal of ATM or the termination of ATM activation”. This is *not* a conclusion of our study. In fact, in the revision, we already stated at the end of discussion about Mre11 “…further experiments are warranted to understand the molecular details of MRE11 function in fork cleavage” (Discussion, second paragraph). We only try to suggest this as a possibility (among others) that could explain the data we saw.

*The second point regards the lack of RAD51 foci in ATMKD cells treated with CPT. The authors attributed this observation to the lack of DSB generation ATMKD cells, rather than impaired RAD51 loading. While this statement is valid with the support the absence (or reduced number) of gH2AX foci and neutral comet assay, the possibility of defective RAD51 loading should not be excluded. The authors stated in the response to reviewer's comment that the ssDNA at stalled replication fork would not lead to RAD51 loading. However, two independent studies have directly or indirectly shown that RAD51 is present at stalled replication forks to promote fork reversal or protect the integrity of nascent DNA strands (Zellweger R et al., JCB, 2015 and Schlacher K et al., Cell, 2011). The later study actually proposed that the stability of RAD51 filament, rather than RAD51 "loading", is important for protecting nascent DNA at stalled fork. Whether RAD51 loading is an issue or not does not affect the core value of this manuscript, but it would be nice if the authors could make a minor revision to include all possibilities.*

We have now cited both references (Zellweger R et al., JCB, 2015 and Schlacher K et al., Cell, 2011) mentioned by the reviewers. We noted that we had already cited Zellweger R et al., JCB, 2015 in our first revision. We now further clarify that we could only exclude a “general” defect that affecting RAD51 loading to “resected DSBs” in *Atm^KD/-^*cells (subsection “ATM-KD blocks replication-dependent removal of Topo-isomerase I DNA adducts at the step of strand cleavage”, second paragraph), since IR induced RAD51 foci formation is *not* significantly reduced in *Atm^KD/-^*cells. We also added discussion that explores the possibility that Rad51 loading or stability at the stalled replication fork might be regulated differently than those occurred at the resected DSBs (from IR) (Discussion, fourth paragraph).